# Highly efficient pure-blue organic light-emitting diodes based on rationally designed heterocyclic phenophosphazinine-containing emitters

Longjiang Xing[1,6], Jianghui Wang[2,6], Wen-Cheng Chen [1,3] ✉, Bo Liu[1], Guowei Chen[1], Xiaofeng Wang[1], Ji-Hua Tan[1], Season Si Chen [4], Jia-Xiong Chen[1,3], Shaomin Ji[1,3], Zujin Zhao [2] ✉, Man-Chung Tang [4] ✉ & Yanping Huo [1,3,5] ✉

Multi-resonance thermally activated delayed fluorophores have been actively studied for high-resolution photonic applications due to their exceptional color purity. However, these compounds encounter challenges associated with the inefficient spin-flip process, compromising device performance. Herein, we report two pure-blue emitters based on an organoboron multi-resonance core, incorporating a conformationally flexible donor, 10-phenyl-5*H*-phenophosphazinine 10-oxide (or sulfide). This design concept selectively modifies the orbital type of high-lying excited states to a charge transfer configuration while simultaneously providing the necessary conformational freedom to enhance the density of excited states without sacrificing color purity. We show that the different embedded phosphorus motifs (phosphine oxide/sulfide) of the donor can finely tune the electronic structure and conformational freedom, resulting in an accelerated spin-flip process through intense spin-vibronic coupling, achieving over a 20-fold increase in the reverse intersystem crossing rate compared to the parent multi-resonance emitter. Utilizing these emitters, we achieve high-performance pure-blue organic light-emitting diodes, showcasing a top-tier external quantum efficiency of 37.6% with reduced efficiency roll-offs. This proposed strategy not only challenges the conventional notion that flexible electron-donors are undesirable for constructing narrowband emitters but also offer a pathway for designing efficient narrow-spectrum blue organic light-emitting diodes.

The pursuit of organic materials capable of emitting narrow-spectrum luminescence in the three primary colors—red, green, and blue (RGB)—is critical for advancing the next generation of organic light-emitting diode (OLED) displays[1–3]. Although thermally activated delayed fluorescence (TADF) materials have garnered attention for achieving 100% exciton utilization, most exhibit broad-spectrum emission due to multiple vibrational transitions[4]. The emergence of boron/nitrogen (B/N)-based polycyclic multi-resonance (MR) TADF materials has

A full list of affiliations appears at the end of the paper. ✉e-mail: wencchen@gdut.edu.cn; mszjzhao@scut.edu.cn; kobetang2021@sz.tsinghua.edu.cn; yphuo@gdut.edu.cn

sparked a paradigm shift in electroluminescence (EL) material design, aiming for narrowband emission and high luminescence efficiency[5,6]. However, a significant hurdle for the development of MR-TADF materials lies in their slow spin−flip process[7–11]. Most MR-TADF structures exhibit comparable locally excited (LE) features in their singlet and triplet excited states[12]. According to El-Sayed's rule[13], the spin−orbit coupling (SOC) matrix element between singlet and triplet states with the same type of orbital is vanishingly tiny, making this spin flipping ($^1LE \rightarrow {}^3LE$) in MR-TADF structures forbidden. This results in ultralow reverse intersystem crossing (RISC) rates ($k_{RISC}$), ranging from $10^3$ to $10^5 s^{-1}$[14], leading to significant triplet accumulation and detrimental exciton losses at high excitation densities.

Modifying MR cores with electron-deficient[15,16] or -rich[17–20] moieties is a proven approach that can markedly boost $k_{RISC}$ by enhancing long-range charge transfer (CT) character in excited states. Nevertheless, because of the unmanageable electron push-pull effect, these methods of CT excited-state regulation often suffer from a series of drawbacks, for instance, decreased photoluminescence quantum yields (PLQYs)[21,22], redshifted emission[18,23,24], deteriorative color purities induced by broadened emission bandwidth[25,26], shoulder peak or tail at long-wavelength regions of emission[17,27]. The negative influences of CT excited state on the luminescence color purity of MR chromophores, especially the pure-blue light-emitting ones, substantially impede their functionalization and further EL performance improvement.

Recently, accelerating spin-flip via combining spin−orbit and vibronic effects, collectively termed the spin−vibronic coupling (SVC), has received much attention[28]. This kind of coupling requires a small energy difference between the states involved, as well as the existence of coupling vibrational modes, which would invoke quantum-mechanically forbidden RISC at non-adiabatic crossings and provide a stepwise spin−flip process with reduced activation energy barrier[29]. Nevertheless, there is only a small fraction of MR-TADF compounds that exhibit close-lying triplet states[30]. Moreover, due to the inherent rigidity of MR chromophores, the suppression of numerous vibronic modes diminishes the efficacy of the SVC process, especially when compared to twisted donor−acceptor (D−A) materials that offer greater conformational freedom[31]. As a result of this constraint, conventional SVC-involved spin−flip in the MR-TADF system remains sporadic and inefficient, leading to $k_{RISC}$ values far $<10^5 s^{-1}$ [32–34].

Herein, we introduce a conformationally flexible-donor-incorporation (CFDI) strategy, combining CT excited-state modulation and SVC as illustrated in Fig. 1 The CFDI strategy involves the deliberate design of conformationally flexible donors, 10-phenyl-*5H*-phenophosphazinine 10-oxide (NPO) and 10-phenyl-*5H*-phenophosphazinine 10-sulfide (NPS), which skillfully avoids $S_1$ state involvement while utilizing electron pull-push effects and multiconformation to optimize the spin-flip process. We validate the CFDI design with two pure-blue MR-TADF emitters, BNCz-NPO and BNCz-NPS, featuring NPO and NPS as functional donors, respectively. Substituting NPO (NPS) selectively switches the triplet locally excited ($^3LE$) character to a charge transfer one ($^3CT$), allowing the spin−flip process. The flexibility of the donors can appropriately render certain conformational freedom that generates dense excited states to realized intense SVC-mediated RISC channels compared to reported SVC tactics. Our findings indicate that while the embedded phosphorus motifs (phosphine oxide/sulfide) finely tune the electronic structure and conformational freedom, the introduction of NPO/NPS only adds vibrational modes in the low-frequency region, thereby preserving the narrowband emission attribute. As a result, we realized a significantly improved $k_{RISC}$ coupled with highly efficient pure-blue luminescence with a peak at 476 nm, a small FWHM of 20 nm, and a superior PLQY approaching unity. Impressively, our champion OLEDs achieved a top-ranking external quantum efficiency (EQE) of 37.6%, representing one of the highest reported efficiencies to date in the field of blue OLEDs.

## Results

### Molecular design, synthesis, and characterization

A prototypical MR framework, BNCz, was employed as the light-emitting core due to its excellent optical properties[35]. Despite several proposed modification strategies to enhance its performance, most BNCz derivatives emit in the blue-green spectrum, presenting a challenge to simultaneously achieving efficient RISC and pure-blue emission. Herein, we introduce an NPO/NPS unit onto the *para*-position of the BNCz skeleton with respect to the B atom, to yield two MR-TADF emitters, namely BNCz-NPO and BNCz-NPS, are depicted in Fig. 1 Compared to previously reported modifying building blocks for MR cores, NPO/NPS serves as a versatile excited-state modifier with distinct advantages. The NPO/NPS unit exhibits moderate-to-weak electron-donating ability due to the negative inductive effect of the embedded P = O/P = S (refer to Supplementary Fig. 1). It is anticipated that the mild CT formed between NPO/NPS and the MR core will compete with the intrinsic short-range CT of the MR structure, leading to blue-shifted emission[36]. In addition, The C-N linking style between BNCz and NPO/NPS is designed to minimize undesired π-bonding features, restricting conjugation extension and bond stretching[37], thereby facilitating narrow-spectrum blue light emission. Moreover, by leveraging the thermal pyramidal inversion behaviors commonly observed in arylphosphines[38], the incorporation of NPO/NPS is expected to enhance the degree of conformational freedom[39]. The larger atomic radius of phosphorus compared to sulfur hints that the NPO and NPS units exhibit multiple metastable conformers similar to phenothiazine[40], further promoting dense excited state alignment and proper vibration modes. Under these circumstances, a significantly improved SVC-mediated RISC process can be realized, given the small energy differences with coupled vibrational modes in such a dense excited-state system with both LE and CT natures.

The synthetic routes to BNCz-NPO and BNCz-NPS are shown in Supplementary Fig. 2. The double Br/Li exchange of Boc-protected bis(2-bromophenyl)amine formed a bis-lithiated species and was then treated with dichloro(phenyl)phosphane to afford dihydrophenophosphazinine. Without further purification, the dihydrophenophosphazinine intermediate was oxidized by $H_2O_2$ or $S_8$ before Boc detachment by trifluoroacetic acid to yield NPO or NPS, respectively. Finally, BNCz-NPO and BNCz-NPS were synthesized using Hartwig-Buchwald C−N coupling reactions of a brominated BNCz derivative with NPO and NPS, respectively. The target materials were purified by column chromatography and temperature-gradient vacuum sublimation and were characterized by $^1H/^{13}C/^{31}P$ NMR, high-resolution mass spectroscopies, and X-ray crystallographic analyses. According to the cyclic voltammetry results (Supplementary Fig. 3), the highest occupied molecular orbital (HOMO) and lowest unoccupied molecular orbital (LUMO) energy levels were calculated to be −5.6 and −3.1 eV for BNCz-NPO and −5.6 and −3.0 eV for BNCz-NPS, respectively. BNCz-NPO and BNCz-NPS demonstrate high decomposition temperatures (5% weight loss) of 503 and 463 °C (Supplementary Fig. 4), respectively, without glass transition points (scanning room temperature to 350 °C), thermally stable for the fabrication of vacuum-deposited OLEDs.

### Crystallographic analysis

The influences of NPO/NPS substituent on molecular conformation and intermolecular interaction are displayed in single-crystal structures. We obtained two distinct types of single crystals for BNCz-NPO and BNCz-NPS, designated as BNCz-NPO-α/BNCz-NPO-β and BNCz-NPS-α/BNCz-NPS-β, by employing an anti-solvent vapor diffusion method with various solvent systems. The selected crystallographic information are summarized in Supplementary Tables 1 and 2. The polymorphism of BNCz-NPO and BNCz-NPS confirms their conformational flexibility under ambient conditions. As shown in Fig. 2, the N−P−C angles are 116.80°, 120.01°, 115.60° and 124.97° for BNCz-

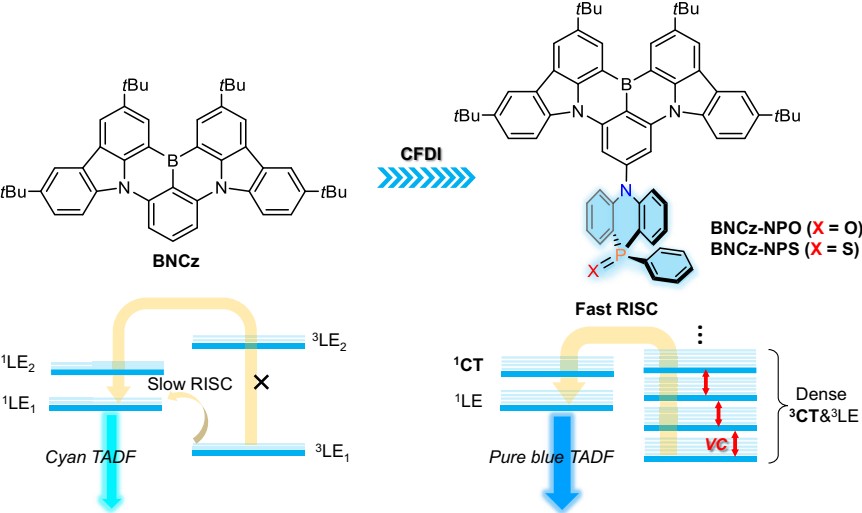

**Fig. 1 | Schematic illustration of molecular design strategy.** The proposed conformationally flexible-donor-incorporation (CFDI) molecular design concept for pure-blue MR-TADF emitters.

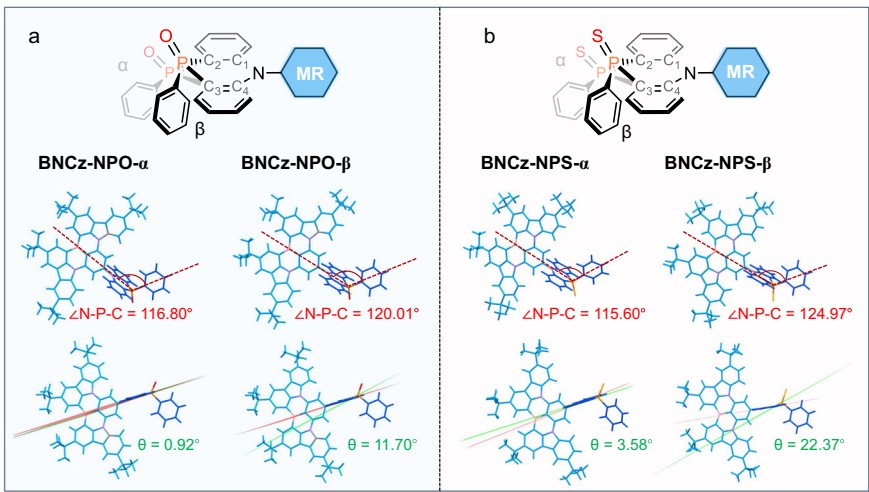

**Fig. 2 | Crystallographic analyses. a** Conformations of BNCz-NPO-α and BNCz-NPO-β. **b** Conformations of BNCz-NPS-α and BNCz-NPS-β.

NPO-α, BNCz-NPO-β, BNCz-NPS-α, and BNCz-NPS-β, respectively. Notably, substantial differences exist in the dihedral angles between the C1–N–C4 and the C2–P–C3 planes ($\theta$). For instance, BNCz-NPO-β exhibits a larger $\theta$ of 11.70° when compared with BNCz-NPO-α (0.92°). As sulfur is more sizable than oxygen, BNCz-NPS is expected to exhibit greater conformational variation than BNCz-NPO. Compared to BNCz-NPO, BNCz-NPS shows larger distortion with $\theta$ of 3.58° and 22.37° for its α and β type crystals, respectively.

The existence of diverse conformers could help to promote multiple excited states and enrich vibration modes in accordance with our design principle. Equally important, unlike phenothiazine derivatives showing a distorted boat-chair conformation[40], the more crowded configuration of phosphine oxide/sulfide imparts a nearly coplanar geometry to the heterocycle skeleton of NPO (NPS). This characteristic may suppress high-amplitude structural deformation and vibration, mitigating potential spectral broadening. Moreover, the bulky NPO/NPS, with significant steric hindrance, induces a nearly orthogonal conformation with an MR-NPO/NPS dihedral angle exceeding 80°, effectively suppressing strong interchromophore interactions (Supplementary Fig. 5). We employed Hirshfeld surface analysis to scrutinize the packing mode in single crystals of BNCz-NPO and BNCz-NPS (Supplementary Fig. 6a). The analysis revealed sparser contact densities and larger contact distances compared to BNCz, indicating weaker intermolecular interactions and reduced interchromophore contacts.

## Theoretical investigation

Theoretical calculations based on density functional theory (DFT) have been conducted to elucidate the electronic structures. Geometry optimizations in the ground states were adapted from crystallographic data. As illustrated in Supplementary Fig. 7, the LUMOs of BNCz-NPO and BNCz-NPS closely resemble that of BNCz, exhibiting a regular distribution in accordance with the MR effect. However, their HOMO distributions exhibit variations between cases. The HOMOs of BNCz-NPO-α, BNCz-NPO-β, and BNCz-NPS-β align with BNCz's HOMO distribution, while that of BNCz-NPS-α delocalizes to the NPS unit. Despite its minimal impact on HOMOs, the electron-donating effect of NPO/NPS causes the HOMO-1 to shift from the MR core to the NPO/NPS units. This underscores the notable influence of conformational changes on electronic configuration imparted by the CFDI strategy.

Figure 3a depicts the flexible potential energy curve scan of BNCz-NPO and BNCz-NPS at the ground state. It is found that a shallow potential well, with an energy difference lower than thermal energy at room temperature ($k_BT \approx 0.026$ eV, $k_B$: Boltzmann constant, $T$: temperature), exists in a wide range of $\theta$ from −40° to 47° for BNCz-NPO.

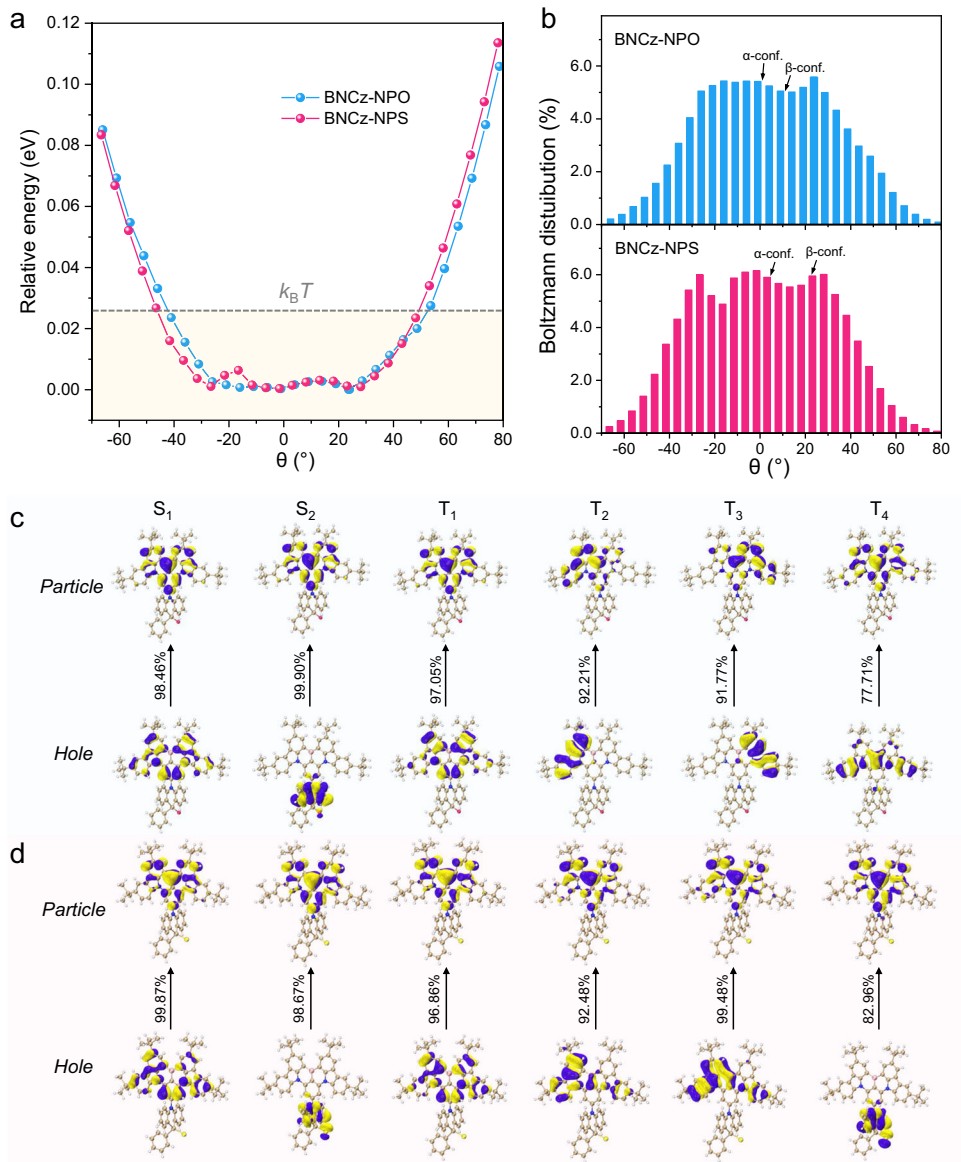

**Fig. 3 | Quantum chemical calculation results. a** Potential energy surface scans of the ground state of BNCz-NPO and BNCz-NPS under vacuum conditions. **b** Boltzmann distributions of the geometries with different dihedral angles at room temperature. Hole-particle distribution for the excited singlet and triplet states of **c** BNCz-NPO and **d** BNCz-NPS.

This range becomes larger from −45° to 50° for BNCz-NPS with higher conformational flexibility. Additionally, we found that the α conformer of BNCz-NPO (BNCz-NPS) exhibits a minimal potential energy difference compared to the β conformer, resulting in a nearly equal Boltzmann distribution ratio between them at room temperature (Fig. 3b). This helps elucidate the observed polymorphism of BNCz-NPO and BNCz-NPS.

Excited state calculations were performed using the spin-component scaling second-order approximate coupled-cluster (SCS-CC2) method with the cc-pVDZ basis set[30]. Supplementary Fig. 8 illustrates a comparatively low $\Delta E_{ST}$ value of ~0.12 eV estimated in vacuum for the compounds, similar to that of BNCz. Natural transition orbital (NTO) analyses were conducted to gain insights into the excited-state nature. Consistent with BNCz (Supplementary Fig. 9), the $S_1$ and $T_1$ states of BNCz-NPO and BNCz-NPS predominantly localize on their MR core (Fig. 3c, d), indicating LE characteristics ($^1$LE and $^3$LE). In contrast, some higher-lying excited states, such as $S_2$, $T_2$, and $T_4$, of BNCz-NPO and BNCz-NPS exhibit long-range CT or hybridized local

and charge-transfer (HLCT) features, while those of BNCz remain dominated by LE. These CT characteristics are quantified by larger values of the distance of charge transfer ($D_{CT}$) and the amount of charge transferred ($q_{CT}$)[41], as shown in Supplementary Table 3. Due to the involvement of CT excited states distinct from the original LE-typed orbital feature of the MR core, both BNCz-NPO and BNCz-NPS display significantly larger SOC matrix elements than BNCz (Supplementary Fig. 10). Notably, BNCz-NPS exhibits larger $D_{CT}$ and $q_{CT}$ values than BNCz-NPO, resulting in larger SOC matrix elements than BNCz-NPO. These SOC values notably surpass those of most MR-type TADF emitters[42].

Excited states with a CT nature are typically more sensitive to the polarity of the environment compared to those with an LE nature. Thus, while BNCz-NPO and BNCz-NPS exhibit comparable excited state energy levels to BNCz due to the weak electron-donating nature of NPO/NPS, their $S_2$, $T_2$, and $T_4$ excited states become more stabilized than those of BNCz in polar $CH_2Cl_2$ (Supplementary Fig. 8). Additionally, due to subtle differences in excited-state energies among

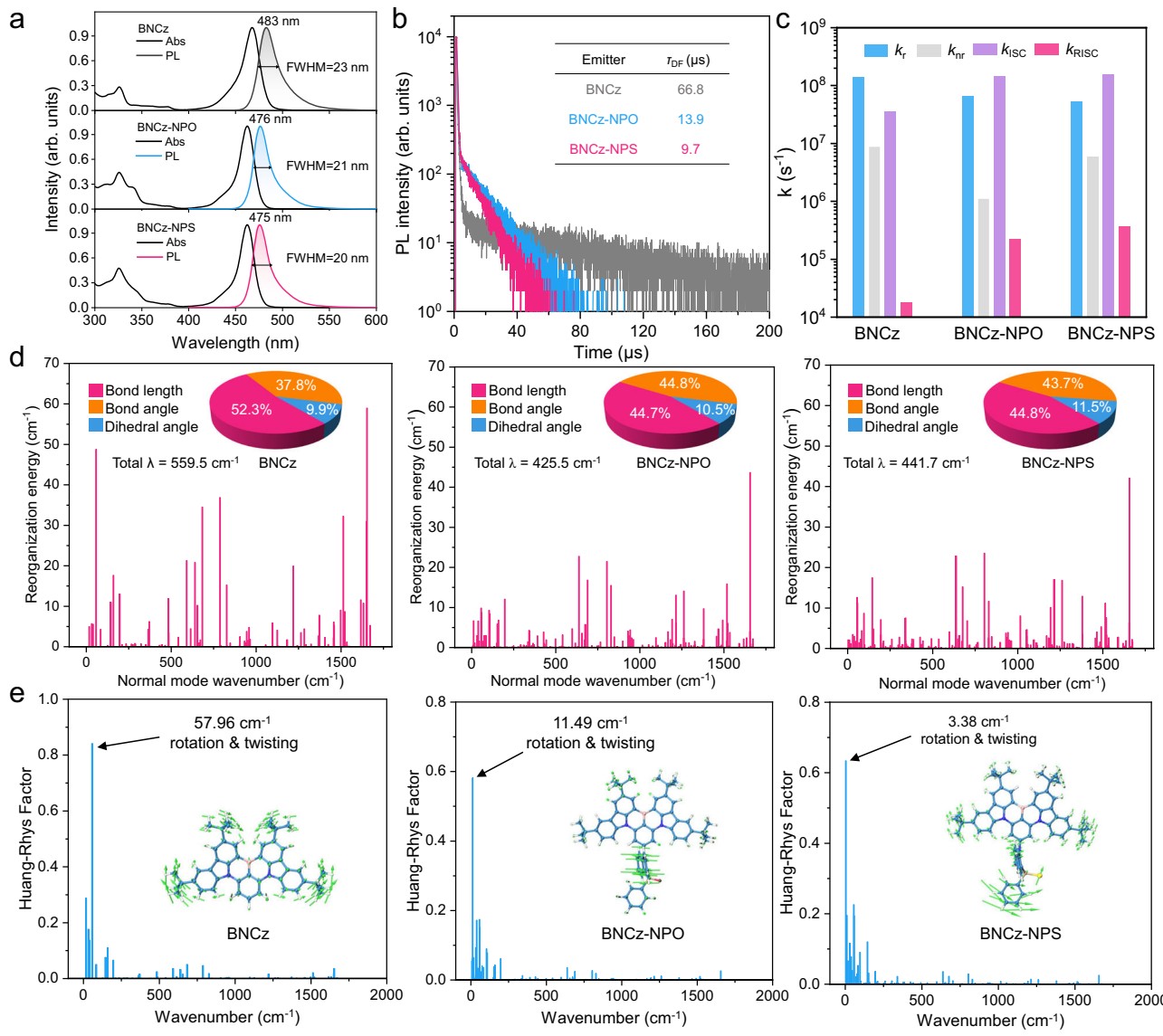

**Fig. 4 | Photophysical characterization and quantum chemical calculations.**
**a** Absorption and PL spectra of BNCz, BNCz-NPO, and BNCz-NPS in toluene
($1 \times 10^{-5}$ M) at room temperature. **b** transient PL decay curves of BNCz, BNCz-NPO, and BNCz-NPS in oxygen-free toluene solution using a variable pulsed laser
($\lambda_{ex} = 375$ nm). **c** Comparison between the calculated rate constants of BNCz, BNCz-NPO, and BNCz-NPS. **d** Reorganization energy versus the normal mode
wavenumbers of BNCz, BNCz-NPO, and BNCz-NPS (insert: pie charts illustrating the
contribution to the total reorganization energy from the bond length, bond angle,
and dihedral angle). **e** Calculated Huang–Rhys factors with values > 0.02 from $S_1$ to
$S_0$ transition for BNCz, BNCz-NPO, and BNCz-NPS (inset: selected vibration modes
with the most significant contribution to the Huang–Rhys factor).

different conformers, BNCz-NPO and BNCz-NPS actually have a much
richer landscape of excited states. Furthermore, the conformationally
flexible NPO/NPS facilitates appropriate vibrational overlap between
the nearly degenerate excited states, promoting reverse internal con-
version. In this scenario, although the spatial orbital occupation (LE) of
the $S_1$ and $T_1$ excitations results in vanishing direct SOC, the close-lying
upper CT (or HLCT) excited states of BNCz-NPO and BNCz-NPS pro-
vide a feasible channel for efficient RISC, wherein higher-order SOC
involving $^3$LE (or $^3$CT) and $^1$CT (or $^1$LE) in thermal equilibrium plays a
crucial role[43].

## Photophysical properties
The absorption and photoluminescence (PL) spectra of BNCz-NPO and
BNCz-NPS in $10^{-5}$ M toluene are shown in Fig. 4a. BNCz-NPO and BNCz-
NPS exhibit intense short-range CT absorption bands peaking at *ca.*
462 nm. Compared to the case of BNCz ($\lambda_{abs} = 468$ nm),

hypochromic shifts in absorption were observed. Following this
trend, BNCz-NPO and BNCz-NPS display shorter-wavelength blue
emissions with a peak at 476 and 475 nm, respectively (BNCz:
$\lambda_{em} = 483$ nm), resulting from the reduced short-range CT character-
istic. The PLQYs of BNCz-NPO and BNCz-NPS in dilute toluene solution
are as high as 98.3% and 91.8%, respectively, which are comparable to
that of BNCz (93.2%). It is worth noting that the PL spectra of the MR-
TADF emitters are unstructured, with a very narrow FWHM of 20 and
21 nm for BNCz-NPO and BNCz-NPS, respectively. Such small FWHM
values were roughly the same as the BNCz MR-core (23 nm) without
spectral shoulders, reflecting that the NPO/NPS subunit does not add
detrimental vibrations. Polarity-sensitive long-range CT character
involvement in the $S_1$ state often impairs emission color purity[18].
Negligible solvatochromism was observed in absorption (Supple-
mentary Fig. 11) and PL spectra (Supplementary Fig. 12), implying
weak long-range CT characters in the ground and $S_1$ states. The emission

peak redshifts from nonpolar $n$-hexane to polar acetonitrile are 17, 13, and 13 nm for BNCz, BNCz-NPO, and BNCz-NPS, respectively (Supplementary Table 4), indicating reduced short-range CT character upon NPO/NPS modification, which is in good accordance with the theoretical results.

From the fluorescence and phosphorescence spectra in toluene measured at 77 K (Supplementary Fig. 13), the $S_1$ and $T_1$ energy levels are estimated to be 2.67/2.52 and 2.64/2.50 eV for BNCz-NPO and BNCz-NPS, respectively. Subsequently, the $\Delta E_{ST}$s of BNCz-NPO and BNCz-NPS are determined to be 0.15 and 0.14 eV, respectively, which align well with the calculation results. These values are sufficiently small to support the exciton upconversion from the $T_1$ to the $S_1$ state, indicative of TADF. PL decays of BNCz, BNCz-NPO and BNCz-NPS consist of ns-scale prompt fluorescence (Supplementary Fig. 14) and µs-scale delayed fluorescence components (Fig. 4b). The corresponding prompt ($\tau_{PF}$) and delayed ($\tau_{DF}$) lifetimes were fitted to be 4.8 ns/13.9 µs for BNCz-NPO and 4.7 ns/9.7 µs for BNCz-NPS, respectively. In contrast, BNCz demonstrated a significantly longer $\tau_{DF}$ of 66.8 µs, indicating a much more efficient RISC process in the presence of a CFDI strategy. Temperature-dependent transient PL measurements (Supplementary Fig. 15) unambiguously confirm the involvement of triplet excitons in light emission through an endothermic RISC process. It is noteworthy that BNCz-NPO and BNCz-NPS exhibit efficient TADF in solution states without the aid of host materials, distinguishing them from most reported MR chromophores with LE-featured singlet and triplet excited states[5,12,44]. The photophysical results demonstrate that the CFDI strategy not only prevents the involvement of NPO/NPS in the $S_1$ states to retain the excellent photophysical properties of the MR core but also simultaneously induces a mild electron push-pull effect to accurately modulate long-range CT features of the high-lying singlet and triplet excited states for an allowed RISC process. The fluorescence radiative decay rate constants ($k_F$) of BNCz-NPO and BNCz-NPS are notably high at $6.5 \times 10^7$ and $5.3 \times 10^7\ \mathrm{s^{-1}}$, respectively (Fig. 4c, Table 1). These values exceed the corresponding $k_{nr}$ values ($1.1 \times 10^6$ and $4.8 \times 10^6\ \mathrm{s^{-1}}$), indicating negligible energy loss during the $S_1 \rightarrow S_0$ transition controlled by the rigid MR core. Importantly, compared to the small $k_{RISC}$ of the prototypical BNCz ($1.8 \times 10^4\ \mathrm{s^{-1}}$), those of BNCz-NPO and BNCz-NPS are increased by 12.2 and 20.5 folds, reaching $2.2 \times 10^5$ and $3.7 \times 10^5\ \mathrm{s^{-1}}$, respectively.

Further investigation of the photophysical properties in doped films (3 wt% in 9-(2-(9-phenyl-9$H$-carbazol-3-yl) phenyl)9$H$–3,9'-bicarbazole, PhCzBCz) is detailed in Supplementary Fig. 16 and Table 1. The BNCz-NPO and BNCz-NPS films exhibit pure-blue emission, peaking at 478 and 481 nm, respectively. The FWHMs are slightly broadened to 26 nm. The doped films do not show broadband emission from excimer or exciplex. The PLQYs of the films are as high as 95.5% and 92.0% for BNCz-NPO and BNCz-NPS, respectively. The $\tau_{PF}$s of BNCz-NPO and BNCz-NPS are 2.8 and 3.0 ns, while their $\tau_{DF}$s are 28.5 and 21.9 µs, respectively. The $k_{RISC}$s of BNCz-NPO and BNCz-NPS in the doped film state were estimated to be $1.6 \times 10^5$ and $2.7 \times 10^5\ \mathrm{s^{-1}}$, respectively. Although these values are slightly lower than those measured in solution due to the reduced effectiveness of the SVC mechanism in the rigid solid state, they still surpass the $k_{RISC}$ values of conventional MR-TADF emitters, which typically fall below $10^5\ \mathrm{s^{-1}}$.

The investigation of photophysical properties for our emitters extended to various doping levels (Supplementary Fig. 17). The findings revealed that BNCz-NPO and BNCz-NPS displayed reduced excimer emission and were less susceptible to concentration-induced emission quenching compared to BNCz. Additionally, the lifetimes of BNCz-NPO and BNCz-NPS exhibited slower decreasing trends with increasing doping concentrations, indicative of their superior solid-state luminescence properties. Furthermore, we investigated the PL behaviors in THF/water mixtures with varying water fractions to highlight the capability to mitigate aggregation-caused quenching (ACQ), as shown in Supplementary Fig. 18. The results showed that

**Table 1 | Photophysical data and kinetic parameters of BNCz, BNCz-NPO, and BNCz-NPS in toluene ($1 \times 10^{-5}$ M) and doped films (3 wt% in PhCzBCz)**

| Emitter | State | $\lambda_{abs}^a$ [nm] | $\lambda_{em}^b$ [nm] | FWHM$^c$ [nm/meV] | $E_{S1}^d$ [eV] | $E_{T1}^d$ [eV] | $\Delta E_{ST}^e$ [eV] | $\Phi_{PF/DF}^f$ [%] | $\tau_{PF}^g$ [ns] | $\tau_{DF}^h$ [µs] | $k_r^i$ [$10^7$ s$^{-1}$] | $k_{nr}^i$ [$10^6$ s$^{-1}$] | $k_{ISC}^i$ [$10^7$ s$^{-1}$] | $k_{RISC}^i$ [$10^5$ s$^{-1}$] |
|---|---|---|---|---|---|---|---|---|---|---|---|---|---|---|
| BNCz | Sol | 468 | 483 | 23/0.13 | 2.62 | 2.52 | 0.10 | 76.4/16.8 | 5.4 | 66.8 | 14.1 | 10.3 | 3.3 | 0.18 |
| | Film | – | 493 | 32/0.17 | 2.61 | 2.52 | 0.09 | 73.9/17.8 | 2.9 | 34.8 | 25.5 | 23.1 | 6.7 | 0.36 |
| BNCz-NPO | Sol | 463 | 476 | 20/0.10 | 2.67 | 2.52 | 0.15 | 31.5/66.8 | 4.8 | 13.9 | 6.5 | 1.1 | 14.2 | 2.2 |
| | Film | – | 478 | 26/0.14 | 2.66 | 2.56 | 0.10 | 20.0/75.5 | 2.8 | 28.5 | 7.1 | 3.3 | 28.2 | 1.6 |
| BNCz-NPS | Sol | 462 | 475 | 21/0.11 | 2.64 | 2.50 | 0.14 | 25.3/66.5 | 4.7 | 9.7 | 5.3 | 4.8 | 15.4 | 3.7 |
| | Film | – | 481 | 26/0.14 | 2.64 | 2.54 | 0.10 | 15.3/76.7 | 3.0 | 21.9 | 5.1 | 4.5 | 27.8 | 2.7 |

$^a$Peak of absorption spectrum.
$^b$Peak of fluorescence spectrum.
$^c$Full width at half-maximum (FWHM) of fluorescence spectrum.
$^d$Lowest excited singlet ($E_S$) and triplet ($E_T$) energies estimated from peaks of the fluorescence and low-temperature phosphorescence spectra recorded at 77 K.
$^e\Delta E_{ST} = E_S - E_T$.
$^f$Absolute photoluminescence quantum yield, fractional quantum yields for prompt fluorescence ($\Phi_{PF}$) and delayed fluorescence ($\Phi_{DF}$).
$^g$Lifetime of prompt fluorescence.
$^h$Lifetime of delayed fluorescence.
$^i$Rate constants of singlet radiative decay ($k_r$), non-radiative decay ($k_{nr}$), intersystem crossing ($k_{ISC}$), reverse intersystem crossing ($k_{RISC}$).

while BNCz exhibited typical ACQ behavior with increasing water fractions, BNCz-NPO and BNCz-NPS maintained stable emission intensities and FWHM values, indicating noteworthy anti-quenching characteristics imparted by the NPO/NPS functionalization. These observations are consistent with the insights gleaned from crystallographic and computational analyses.

It is widely recognized that introducing flexible units in a conjugated emitter typically results in pronounced structural deformation during electronic transitions, leading to spectral broadening and reduced luminescence efficiency[45]. Therefore, it is intriguing to observe that incorporating the flexible NPS/NPS moiety into the MR chromophore manages to preserve the merits of narrowband emission and a high PLQY. Structural changes during the $S_1 \rightarrow S_0$ transitions were evaluated using root-mean-square deviations (RMSDs). The RMSDs for BNCz-NPO and BNCz-NPS were calculated to be 0.093 and 0.139 Å, respectively, compared to 0.084 Å for BNCz (Supplementary Fig. 19). As anticipated, the larger RMSD values originate from the fluctuation of the flexible NPO and NPS groups at the peripheral terminal. Quantitative analysis of the intramolecular motions of BNCz, BNCz-NPO, and BNCz-NPS has been conducted through reorganization energy calculation (Fig. 4d). Surprisingly, BNCz-NPO and BNCz-NPS exhibit remarkably small total reorganization energies of 425.5 and 441.7 cm$^{-1}$, respectively, even surpassing the value for BNCz without NPO/NPS (559.5 cm$^{-1}$). This indicates that the CFDI strategy may increase the number of vibration modes, but the effect on the total reorganization energies is limited since the triphenylphosphine backbone of NPO/NPS is immobilized by a C−N−C locking and strong intramolecular motions of Ph-P = O(S) units (refer also to Supplementary Fig. 19). The flexibility introduced by the NPO/NPS modification increases the share of low-frequency modes related to bond angle changes from 37.8% (BNCz) to 44.8%/43.7%. In contrast, high-frequency modes associated with changes in bond length, which are closely linked to spectral broadening and the appearance of shoulder peaks, are effectively suppressed.

The emission spectra of BNCz-NPO and BNCz-NPS were simulated by the Frank-Condon analysis for the $S_1 \rightarrow S_0$ transition, and Huang−Rhys factors (S) of the vibrational modes were calculated to elucidate the spectral progression. The simulated emission wavelengths and profiles are in accordance with the experimental results (Supplementary Fig. 20). The principal vibrational modes of BNCz-NPO and BNCz-NPS are identified at frequencies of 11.49 and 3.38 cm$^{-1}$, respectively, primarily arising from the twisting and rotation vibrations of the NPO and NPS peripheral units. A more detailed inspection might even suggest that the additional vibrations introduced by NPO/NPS are predominantly located in the low-frequency region with wavenumbers less than 250 cm$^{-1}$, while the high-frequency vibrational modes are limited (refer to Supplementary Fig. 21 and Supplementary Table 5). The restrained high-frequency stretching vibrations, coupled with the structural reorganization between $S_0$ and $S_1$, contribute to the preservation of small overall reorganization energies responsible for the ultrasmall FWHM values. It has been emphasized that concurrently enhancing low-frequency vibrations and reducing high-frequency vibrations is pivotal for achieving narrow-spectrum emission in organic emitters[45]. The experimental and calculation results collectively illustrate that the CFDI strategy can not only enrich the excited states and vibration modes for realizing efficient SVC-mediated RISC but also retain highly efficient narrow-spectrum emissions.

### Electroluminescence properties

To evaluate EL properties of the proposed emitters, a set of OLEDs were first prepared with a structure of indium tin oxide (ITO)/dipyrazino[2,3-f:2′,3′-h] quinoxaline-2,3,6,7,10,11-hexacarbonitrile (HATCN, 5 nm)/4,4′-cyclohexylidenebis[N,N-bis(p-tolyl)aniline] (TAPC, 50 nm)/ 4,4′,4″-tris(carbazol-9-yi)triphenylamine (TCTA, 5 nm)/1,3-bis(carbazol-9-yl)benzene (mCP, 5 nm)/ 1-5 wt% BNCz-NPO or BNCz-NPS:

PhCzBCz (EML, 20 nm)/2,8-bis(diphenyl-phosphoryl)-dibenzo[b,d] furan (PPF, 5 nm)/1,3,5-tri[(3-pyridyl)-phen-3-yl]benzene (TmPyPB, 30 nm)/LiF (1 nm)/Al (120 nm), as displayed in Fig. 5a. The EL performances are depicted in Fig. 5, and relevant key parameters are summarized in Table 2. The 1 wt% doped OLEDs based on BNCz-NPO and BNCz-NPS, respectively, exhibit pure-blue emission at 478 and 474 nm with an FWHM of ~26 nm (Fig. 5b), and the corresponding Commission Internationale de I'Éclairage (CIE) coordinates are (0.11, 0.17) and (0.12, 0.14). This human-friendly blue light, with minimal photon energy in wavelengths <455 nm, aligns with the Bio-Blue display concept proposed by Samsung[46], reducing the risk of retinal damage (blue light hazard)[47].

All devices exhibited an onset voltage of ~3.2 V ($V_{on}$ at 1 cd m$^{-2}$), indicative of efficient carrier injection and transport (Fig. 5c). A marginal redshift in the emission maxima was observed with an increase in doping concentration, while the FWHM remained nearly unchanged, owing to the bulky orthogonal molecular geometries of BNCz-NPO and BNCz-NPS. Both emitters achieved their optimal EL performance at a 3 wt% doping concentration. The EQE$_{max}$ of the BNCz-NPO-based device is as high as 32.1%, while that of the BNCz-NPS-based device is 29.6% (Fig. 5d). The EQE$_{max}$ of 32.1% is among the highest values for all non-sensitized blue MR-TADF OLEDs[48–50]. More importantly, these OLEDs exhibit much-reduced efficiency roll-offs under high exciton density. The EQEs of BNCz-NPO and BNCz-NPS maintain a high level of 21.7% and 23.5% (EQE$_{100}$) at a display relevant luminance of 100 cd m$^{-2}$, over 12.9% of BNCz without functionalization[51]. The alleviated efficiency roll-offs in our devices can be attributed to the enhanced RISC of BNCz-NPO and BNCz-NPS, effectively mitigating triplet-involved exciton quenching. Supplementary Fig. 22 illustrates the EL stabilities of BNCz-NPO and BNCz-NPS. We observed that the BNCz-NPO-based OLED exhibits a significantly prolonged $T_{50}$ (defined as the time when the brightness diminishes to half of its initial value), defined as the time when the luminance diminishes to half of its initial value, surpassing three-fold that of the BNCz-NPS-based device (15.2 h compared to 3.0 h). We speculate that the lower EL stability of BNCz-NPS could be attributed to the weaker bonding of the P = S group in its molecular structure.

Hyperfluorescence (HF) OLEDs using 2,3,4,5,6-pentakis-(3,6-di-tert-butyl-9H-carbazol-9-yl) benzonitrile (5TCzBN) as a TADF sensitizer in EML were fabricated to further optimize the EL performances[52]. Great overlap between the PL of 5TCzBN and absorption of BNCz-NPO/ BNCz-NPS guarantees efficient Förster energy transfer (Supplementary Fig. 23). The OLEDs were constructed with the configuration of ITO/HATCN (5 nm)/TAPC (50 nm)/TCTA (5 nm)/mCP (5 nm)/3 wt% BNCz-NPO or BNCz-NPS: 10–20 wt% 5TCzBN: PhCzBCz (20 nm)/PPF (5 nm)/TmPyPB (30 nm)/LiF (1 nm)/Al (120 nm) (HF-I type). The EL characteristics are depicted in Fig. 6a and Supplementary Fig. 24, with key parameters summarized in Table 2 and Supplementary Table 6. The optimal concentration of 5TCzBN is 10 wt%. Compared with the non-sensitized devices, the efficiency roll-offs of the devices are significantly suppressed. Impressively, the EQE retained as high as 31.6% and 26.4% at high luminance of 100 and 1000 cd cm$^{-2}$, respectively, for BNCz-NPS (Fig. 6a). Though the efficiency roll-offs are significantly suppressed, the improvement of maximum EQE in HF-I OLEDs are not obvious. Thus, further optimization was performed using an interlayer sensitization structure[53] with a more efficient TADF sensitizer, 9-(5′-(4,6-diphenyl-1,3,5-triazin-2-yl) [1,1′:3′,1″-terphenyl]−2′-yl)−3,6-diphenyl-9H-carbazole (PPCz-Trz)[54]. Another set of HF OLEDs was fabricated with a device configuration of ITO/HATCN (5 nm)/TAPC (50 nm)/TCTA (5 nm)/3 wt% BNCz-NPO (BNCz-NPS): PhCzBCz (10 nm)/10 wt% PPCz-TRZ: PPF (2 nm)/PPF (5 nm)/TmPyPB (30 nm)/LiF (1 nm)/Al (120 nm) (HF-II). The device structure and characteristics are shown in Supplementary Fig. 25. In HF-II OLEDs, the PPCz-Trz sensitizer was doped into PPF with high polarity to harvest exciton energy more efficiently, thereby enhancing sensitizing efficiency through a long-range Förster

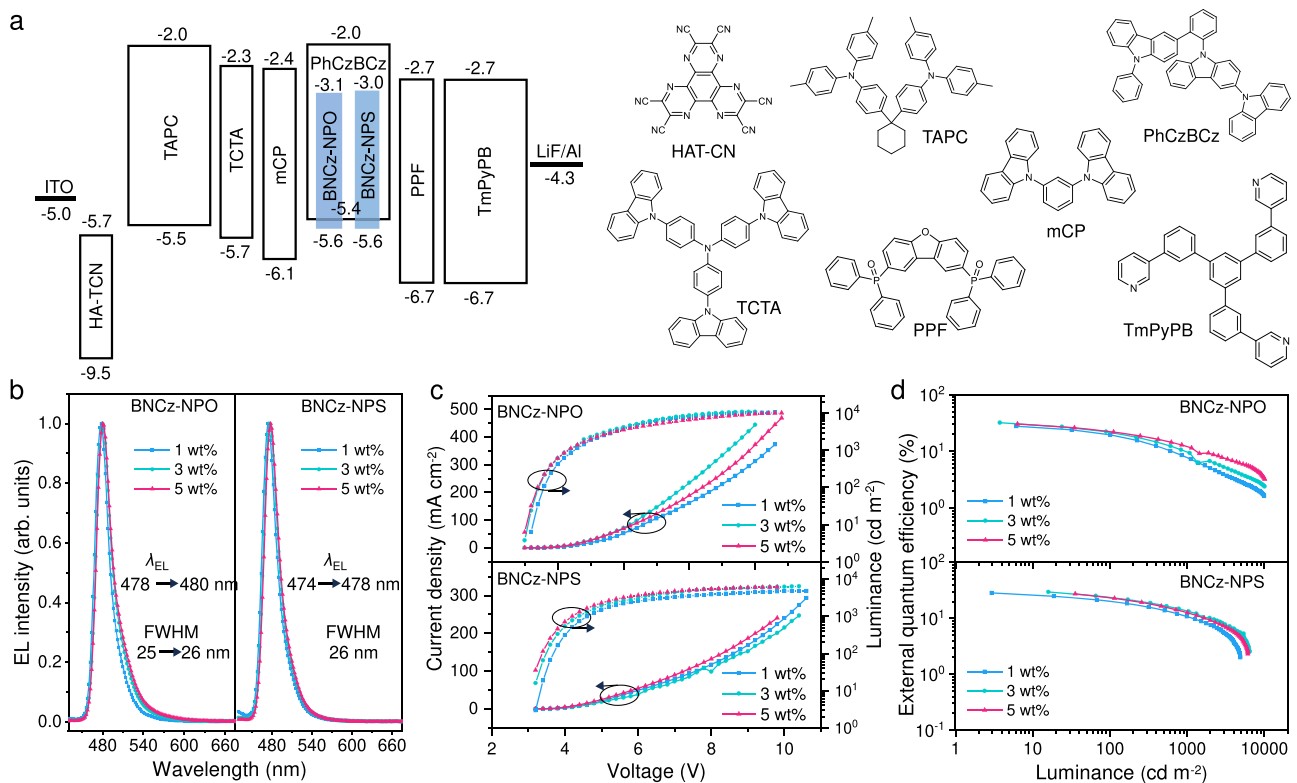

**Fig. 5 | EL performances of non-sensitized OLEDs. a** Device architecture, energy diagram, and functional layers of the OLEDs. **b** EL spectra measured at 100 cd m$^{-2}$. **c** Current density–voltage–luminance characteristics. **d** EQE–luminance curves.

**Table 2 | Summary of the EL performances of BNCz-NPO and BNCz-NPS**

| Emitter | Conc. [wt%] | $V_{on}$[a] [V] | $L_{max}$[b] [cd m$^{-2}$] | CE[c] [cd A$^{-1}$] | PE[d] [lm W$^{-1}$] | EQE[e] [%] | $\lambda_{EL}$[f] [nm] | FWHM[g] [nm] | CIE[h] [x, y] |
|---|---|---|---|---|---|---|---|---|---|
| BNCz-NPO | 1 | 3.2 | 10,510 | 32.6/23.8/13.0 | 32.0/21.0/9.1 | 30.4/22.2/12.1 | 478 | 25 | (0.11, 0.17) |
| BNCz-NPO | 3 | 3.0 | 10,960 | 43.0/28.5/13.1 | 45.0/26.4/9.5 | 32.1/21.7/9.7 | 480 | 26 | (0.11, 0.22) |
| BNCz-NPO | 5 | 3.0 | 10,170 | 40.5/28.4/11.3 | 42.4/26.3/8.3 | 27.8/19.5/7.8 | 480 | 26 | (0.11, 0.25) |
| BNCz-NPO (HF-I) | 3 | 2.8 | 22,660 | 40.6/37.5/27.6 | 45.5/37.7/23.3 | 30.2/28.0/20.7 | 480 | 26 | (0.12, 0.22) |
| BNCz-NPO (HF-II) | 3 | 2.8 | 13,850 | 44.7/36.2/14.1 | 50.1/37.4/12.7 | 37.6/30.2/12.0 | 480 | 26 | (0.11, 0.18) |
| BNCz-NPS | 1 | 3.2 | 4906 | 23.7/16.7/9.0 | 23.3/14.3/6.1 | 28.5/20.0/10.8 | 474 | 26 | (0.12, 0.14) |
| BNCz-NPS | 3 | 3.2 | 6491 | 30.6/24.4/14.0 | 30.0/22.0/10.1 | 29.6/23.5/13.5 | 476 | 26 | (0.11, 0.16) |
| BNCz-NPS | 5 | 3.2 | 6154 | 30.3/25.8/14.1 | 29.7/23.9/10.4 | 27.4/23.3/12.6 | 478 | 26 | (0.11, 0.17) |
| BNCz-NPS (HF-I) | 3 | 3.0 | 26,330 | 50.1/49.4/41.5 | 52.4/50.3/36.1 | 31.9/31.6/26.4 | 480 | 30 | (0.14, 0.24) |
| BNCz-NPS (HF-II) | 3 | 3.0 | 14,640 | 40.2/37.0/21.9 | 39.5/34.4/17.4 | 32.2/29.2/17.3 | 476 | 30 | (0.13, 0.18) |

[a]Voltage measured at 1 cd m$^{-2}$.
[b]Maximum luminance.
[c]Maximum current efficiency at maximum, 100, 1000 cd m$^{-2}$, respectively.
[d]Maximum power efficiency at maximum, 100, 1000 cd m$^{-2}$, respectively.
[e]External quantum efficiency at maximum, 100, 1000 cd m$^{-2}$, respectively.
[f]EL peak.
[g]Full width at half maximum of EL spectrum.
[h]Recorded at 4 V.

energy transfer process[55]. As shown in Fig. 6b and Table 2, an ultrahigh EQE$_{max}$ (EQE$_{100}$) was achieved with a value of 37.6% (30.2%) for the BNCz-NPO-based HF-II device with the corresponding CIE coordinates of (0.11, 0.18) and with a narrow FWHM of 26 nm.

The EL performances of representative narrow-spectrum blue MR-TADF emitters are summarized in Fig. 6c and Supplementary Table 7. Notably, the highest efficiencies of narrow-spectrum blue OLEDs are based on multiple-boron MR-TADF motifs that often need

intricate synthetic procedures, and only a small fraction of MR-TADF emitters meet the requirements of pure-blue narrow-spectrum emission (CIE$_y$ < 0.25, FWHM < 40 nm) and high efficiency (EQE > 30%) at the same time. The molecular design presented here, involving a simple conformationally flexible donor modulation on a mono-boron MR core, allows our MR-TADF emitters to fulfill these demands, enabling the development of robust pure-blue OLEDs. Moreover, BNCz-NPO OLEDs achieve record-setting efficiencies among mono-

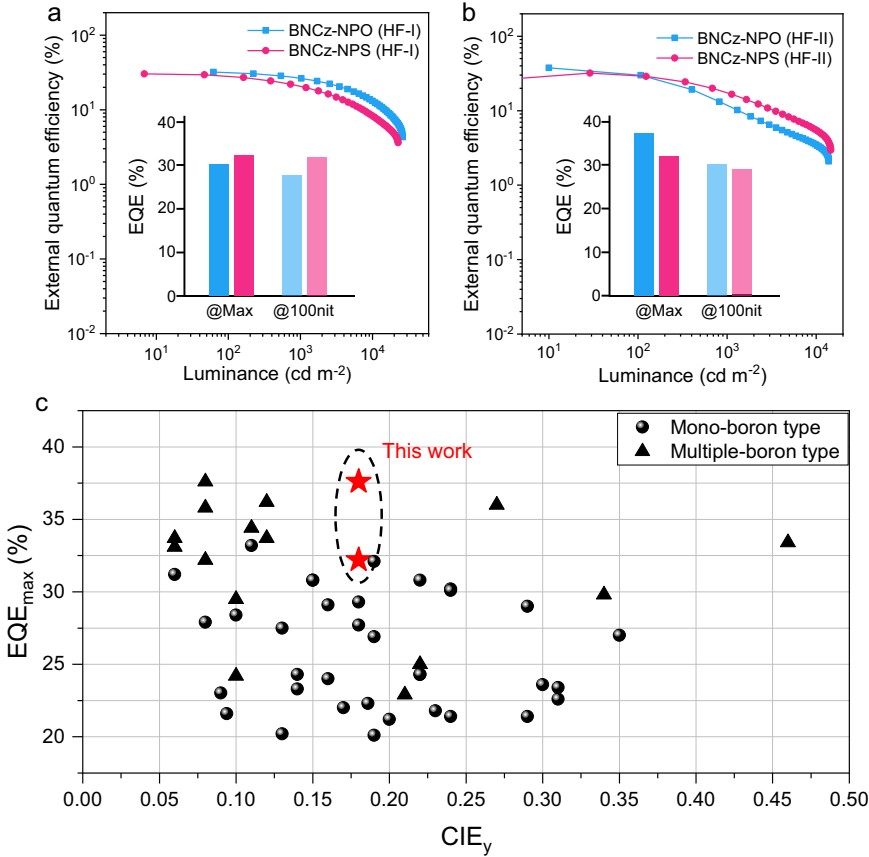

**Fig. 6 | EL performances of HF OLEDs.** EQE-luminance characteristics of **a** HF-I and **b** HF-II OLEDs. **c** EQE$_{max}$ comparison in terms of CIE$_y$ value to reported blue MR-TADF OLEDs with FWHM < 40 nm; circle and triangle represent the EL data for mono- and multiple-boron emitters, respectively (references for the plotted data are given in Supplementary Table 7).

boron MR-TADF OLEDs and prove competitive with the most efficient multiple-boron counterparts[56,57].

## Discussion

In summary, this study introduces two blue MR-TADF materials, BNCz-NPO and BNCz-NPS, meticulously designed through the CFDI strategy. Through extensive photophysical and theoretical analyses, we have revealed that the incorporation of the medium-to-weak electron-donating NPO (NPS) unit into the MR core selectively tunes the high-lying $^3$LE states to $^3$CT ones, enabling allowed RISC, while the narrow-spectrum emissive $S_1 \rightarrow S_0$ transition can be perfectly retained. Crucially, this strategy imparts appropriate conformational freedom to the MR-TADF molecules, enriching dense excited states and their vibrational coupling, thereby facilitating RISC via the SVC mechanism. As a result, BNCz-NPO and BNCz-NPS exhibit significantly improved RISC, with $k_{RISC}$ values of $2.2 \times 10^5$ and $3.7 \times 10^5$ s$^{-1}$, respectively, compared to the NPO/NPS-free parent molecule. A device based on BNCz-NPO demonstrates highly efficient pure-blue emission, peaking at 476 nm with a narrow FWHM of 20 nm and a PLQY approaching unity. The pure-blue OLED based on BNCz-NPO achieves a top-ranking EQE of 37.6%, coupled with a reduced efficiency roll-off.

This work underscores the efficacy of the CFDI strategy in simultaneously preserving narrow-spectrum characteristics and improving RISC rates for MR-TADF emitters. Typically, flexible building blocks are avoided in constructing narrowband emitters due to the potential for inducing violent structural deformation and spectral broadening. The proposed CFDI strategy challenges this conventional notion, opening an avenue toward the realization of high-performance, narrow-spectrum blue OLEDs. Additionally, the CFDI strategy may offer a viable approach to enhancing the film-forming properties of rigid, planar MR-TADF emitters, extending their application in large-area solution-processed manufacturing.

## Methods

### Characterization of the MR-TADF compounds

$^1$H, $^{13}$C, $^{31}$P NMR measurements were recorded with a Bruker AVANCE III HD-400 NMR spectrometer with chemical shifts reported relative to tetramethylsilane ($\delta = 0$ ppm). High-resolution mass spectra (HRMS) were recorded by the Electrospray Ionization (ESI) method with a Thermo Scientific Q Exactive instrument.

### X-ray crystallography

Single crystals were grown in CH$_2$Cl$_2$/$n$-hexane or CHCl$_3$/MeOH mixtures, and the crystallographic data were collected at 100, 150, or 170 K on a Rigaku Oxford Diffraction Supernova Dual Source diffractometer equipped with an AtlasS2 CCD using Cu Kα radiation. Data reduction was carried out with the diffractometer's software. The structures were solved by direct methods using Olex2 software, and the non-hydrogen atoms were located from the trial structure and then refined anisotropically with SHELXL-2014 using a full-matrix least-squares procedure based on F$^2$. The weighted $R$ factor, w$R$ and goodness-of-fit $S$ values were obtained based on F$^2$. The hydrogen atom positions were fixed geometrically at the calculated distances and allowed to ride on their parent atoms.

### Thermal properties

A STA409PC Thermogravimetric Analyzer and a NETZSCH thermal analyzer (DSC 204 F1) were used to measure the decomposition

temperature (5% weight loss, $T_d$) and glass transition temperature ($T_g$), respectively. The tests were performed at a heating rate of 10 °C min$^{-1}$ under an $N_2$ atmosphere.

## Photophysical measurements

The UV-vis absorption spectra were recorded by Shimadzu UV-2700 UV–vis spectrophotometer equipped with a xenon flash lamp. Steady-state and time-resolved PL spectra were recorded with an Edinburgh FLS980 spectrophotometer. Temperature-dependent measurements were conducted within an Optistat DN optical cryostate (Oxford Instruments). The prompt and delayed transient PL decay profiles were performed using a picosecond pulsed diode laser (EPL-375) and a variable pulse length diode laser (VPL-375) as excitation sources, respectively. Absolute PLQYs were measured with an integrating sphere incorporated into the FLS980 spectrofluorometer.

## Cyclic voltammetry measurements and HOMO/LUMO determination

Cyclic voltammetry (CV) was recorded in acetonitrile of tetra-*n*-butylammoniumhexafluorophosphate (Bu$_4$NPF$_6$) (0.1 M) with a scan rate of 100 mV s$^{-1}$ in an Autolab PGSTAT302N electrochemical workstation using a three-electrode cell (Ag/AgNO$_3$ reference electrode, Pt wire counter electrode, and glassy carbon working electrode). Fc/Fc$^+$ (0.18 eV against Ag/AgNO$_3$) was used as an external standard for calibrating the reference electrode. The energy level of HOMO and LUMO were calculated according to the equation of [HOMO = −($E_{onset}$ + 4.8) eV] and [LUMO = (HOMO + $E_g$) eV], where $E_{onset}$ is the onset oxidation potentials, and $E_g$ is the optical bandgap obtained from the absorption onset.

## OLED fabrication and measurements

The glass substrates precoated with a 90-nm layer of ITO with a sheet resistance of 15–20 Ω per square was successively cleaned in an ultrasonic bath of acetone, isopropanol, detergent, and deionized water, respectively, taking 10 min for each step. Then, the substrates were dried in a 70 °C oven. Before the fabrication processes, to improve the hole injection ability of ITO, the substrates were treated with O$_2$ plasma for 10 min. The vacuum-deposited OLEDs were fabricated under a pressure of <5 × 10$^{-4}$ Pa in the Suzhou Fangsheng OMV-FS380 vacuum deposition system. Organic materials, LiF and Al were deposited at rates of 1–2, 0.1, and 5 A s$^{-1}$, respectively. The effective emitting area of the devices was 9 mm$^2$, determined by the overlap between the anode and cathode. The luminance–voltage–current density and external quantum efficiency were characterized with a dual-channel Keithley 2614B source meter and a PIN-25D silicon photodiode. The EL spectra were obtained via an Ocean Optics USB 2000+ spectrometer with a Keithley 2614B source meter. All the characterizations were conducted at room temperature in ambient conditions without any encapsulation, as soon as the devices were fabricated.

## Data availability

All the data and methods are present in the main text and the supplementary materials. The X-ray crystallographic data for structures reported in this study have been deposited at the Cambridge Crystallographic Data Center (CCDC), under deposition numbers CCDC 2253324 (BNCz-NPO-α), 2282192 (BNCz-NPO-β), 2285218 (BNCz-NPS-α) and 2247937 (BNCz-NPS-β). Copies of the data can be obtained free of charge via https://www.ccdc.cam.ac.uk/structures/. Source data are provided with this paper.

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

## Acknowledgements

This work was supported by the National Natural Science Foundation of China (Project Nos. U2001222 for Y.H., U23A20594 for Y.H. and Z.Z., U22A20399 for W.-C.C., 22375066 for Z.Z., 22275114 for M.-C.T.), Guangdong Basic and Applied Basic Research Foundation (Project Nos. 2023B1515040003 for Z.Z. and 2022B1515020041 for M.-C.T.), Science and Technology Planning Project of Shenzhen Municipality (Project No: WDZC20220817160017003 for M.-C.T.).

## Author contributions

W.-C.C. and Y.H. initiated and designed the research. W.-C.C. and L.X. designed the MR-TADF compounds. L.X. synthesized the compounds. L.X., B.L., G.C., X.W. J.-X.C., S.J. conducted the characterization, and photophysical measurements of the compounds. J.W. and Z.Z. carried out the OLED fabrication and characterizations. L.X., B.L. J.-H.T., and S.S.C. performed and analyzed the computational calculations. W.-C.C., Z.Z., M.-C.T., and Y.H. supervised the work. All authors discussed the results and contributed to the manuscript.

## Competing interests

The authors declare no competing interests.

## Additional information

[1]School of Chemical Engineering and Light Industry, Guangdong University of Technology Guangzhou, 510006 Guangzhou, P. R. China. [2]State Key Laboratory of Luminescent Materials and Devices, Key Laboratory of Luminescence from Molecular Aggregates of Guangdong Province, South China University of Technology, 510640 Guangzhou, P. R. China. [3]Guangdong Provincial Laboratory of Chemistry and Fine Chemical Engineering Jieyang Center, 515200 Jieyang, P. R. China. [4]Institute of Materials Research, Tsinghua Shenzhen International Graduate School, Tsinghua University, 518055 Shenzhen, P. R. China. [5]Analytical & Testing Center, Guangdong University of Technology, 510006 Guangzhou, P. R. China. [6]These authors contributed equally: Longjiang Xing, Jianghui Wang. ✉e-mail: wencchen@gdut.edu.cn; mszjzhao@scut.edu.cn; kobetang2021@sz.tsinghua.edu.cn; yphuo@gdut.edu.cn

