## [Peer Review File · Nature Communications]

Highly efficient pure-blue organic light-emitting diodes based on rationally designed heterocyclic phenophosphazinine-containing emittersREVIEWER COMMENTS

Reviewer #1 (Remarks to the Author):

In this paper, chemical modification of the MR emitter was tried using donating group. The concept of the paper is not novel because the same donor substitution was already tried in several papers. Although detailed analysis of the materials was carried out, the novelty of the material design concept is relatively low.

Comments:

Regarding the reference material BNCz, comparisons were made in some analyses, while other analyses did not include a comparative analysis. To better understand the improvements in the synthesized material, please provide a supplementary analysis of the areas that are not compared to promote a more effective understanding.

Theoretical investigation

1. In Figure 3, the analysis of the S_n and T_n states in the excited state suggested that the RISC process could be improved. However, it should be incorporated the calculation results for the reference material to identify any differences from the reference material and elucidate them in the main text.

In addition, the DFT analysis indicates a substantial difference of 0.45 eV between S_1 and T_1 , which was significantly larger than the experimental value of 0.15 eV. Therefore, it is uncertain whether the SOC calculation results for the S_1 and T_n states asserted in Figure 3 are reasonable. Considering the ref.29 you mentioned in the introduction (Page 4, line 74), employing basis sets such as “SCS-CC2” in the DFT calculations could potentially yield more suitable results for MR-type materials.

(Ref. 29: J. Chem. Theory Comput. 2022, 593, 4903.)

Photophysical properties

2. In Figure 4, a comprehensive comparative analysis with the reference material BNCZ facilitated a clear identification of improvements in the synthesized material. It is recommended to extend this comparison to the transient PL data in Figure 4b, providing an analysis of BNCZ's results for a more thorough understanding of the synthesized material.

3. Transient PL data in Figure. 4b and the calculated constant rate in Table 1 fill in information about the solution state, while excepting the solid-state result to supporting information. Since the solid-state result tends to be more similar to the actual device, please transfer this data to the main text.

4. Page 12. From line 270. “Importantly, compared to the small kRISC of the prototypical BNCz ($1.34 \times 10^4 \text{ s}^{-1}$),⁴² those of BNCz-NPO and BNCz-NPS are markedly increased by 16.8 and 27.7 folds, reaching 2.25×10^5 and $3.71 \times 10^5 \text{ s}^{-1}$, respectively.”

☐ The RISC rate constant of Ref. 42 is the result of the film state calculation, and the value of the synthesized material is compared and analyzed with the solution state. Please refer to the results analyzed in the same analysis condition to correct the degree of improvement in the RISC rate.

5. As you mentioned in introduction part, the addition of an additional donor group to the MR core as a way to improve RISC rate may lead to a decrease in PLQY (Page 3. From line 62), so please compare and mention the results of PLQY analysis between the synthesized material and the reference material in the main text.

6. Page 11. From line 250

☐ There are only ΔE_{ST} values in the main text and table, and no S1, T1 data. For intuitive confirmation, please add singlet, triplet energy values.

Crystallographic analysis

7. Page 7. From line 161. “Moreover, the bulky NPO/NPS, with significant steric hindrance, induces a nearly orthogonal conformation with an MR-NPO/NPS dihedral angle exceeding 80° effectively suppressing strong interchromophore interactions (Figure S4).”

☐ As a result of single crystal analysis, it was said that the introduction of NPO/NPS units into the core would inhibit the intermolecular interaction. In order to confirm the characteristics of the unit, please analysis and explain how effectively excimer and concentration quenching were suppressed through analysis such as solid-PL, transient PL, PLQY according to the doping concentration along with the reference core.

Electroluminescence Properties

8. Page 15. From line 376.

☐ In the Hyperfluorescence OLED system, the introduction of an interlayer sensitization structure was mentioned to suppress Dexter energy transfer and further optimize the system. However, it would be beneficial to provide more detailed quantification or explanation, such as specific values related to exciton lifetimes or the degree to which Dexter energy transfer is inhibited for a more understanding.

9. Furthermore, the authors should provide device lifetime results as a benchmark for future research.

Reviewer #2 (Remarks to the Author):

In this manuscript, the authors proposed a conformation-flexible-donor-incorporation (CFDI) strategy, which can simultaneously regulate the long-range charge transfer (CT) character and spin-vibronic coupling (SVC) of molecules. Based on this strategy, two efficient blue luminescent materials, BNCz-NPO and BNCz-NPS, were designed based on a MR-TADF molecule BNCz by introducing NPO and NPS as functional donors. Surprisingly, the transition characteristics of the S1 and T1 states of the molecules do not be changed after the introduction of functional donor, and NPO and NPS do not generate more vibration modes in the low frequency region. As a result, the fluorescence spectra were not broadened. The FWHM values of BNCz-NPO and BNCz-NPS in toluene are only 20 nm and 21 nm. The functional donors NPO and NPS equip the higher energy excited states (S2, T2, etc.) with CT nature, which allows the spin flipping process. Moreover, the flexible structure of the donor group increases the number of triple excited state nearby the S1 state, which increases the intense SVC-mediated RISC channels and greatly improves the kRISC value. The device employing BNCz-NPO (BNCz-NPS) as the emitter exhibited a pure blue emission with an FWHM of 26 nm (30 nm) and an EQEmax of 37.6% (32.2%). These results are excellent. The manuscript is recommended for publication with minor revisions presented below.

1. The single crystal analysis of BNCz-NPO and BNCz-NPS indicated two stable configurations in both crystals. I wonder which configuration the molecules tend to stay in solution or film. Can the two configurations transform each other?
2. The PLQYs of BNCz-NPO and BNCz-NPS in toluene are given in Table 1. Please also give the PLQYs of fluorescence and TADF components.
3. The energy levels and NTOs of Sn and Tm states are given in Figure 3. However, the NTO analysis should involve a transition from one orbital to another, such as NTO No. 15 to NTO No. 16, rather than showing only one NTO in the figure. Additionally, the BNCz molecule is used as a reference in the paper. Please provide the calculation results of BNCz, including the energy-levels and NTOs of the Sn and Tm, and the SOC matrix elements between the Tm and Sn.

Reviewer #3 (Remarks to the Author):

This manuscript presents a design strategy for blue MR-TADF materials by incorporating conformationally flexible donor (NPO and NPS) moieties to enhance the spin-flip of the molecules while maintaining narrowband emission. The novel and effective molecular design make these two molecules exhibit outstanding device performance in both non-sensitized and sensitized devices, placing the molecules at the forefront of the MR-TADF field. The proposed design strategy challenges conventional thinking by effectively integrating flexible and rigid groups, offering readers a fresh approach to molecular design. I recommend publication of the work in Nature Communications after minor revisions as outlined below:

1. There have been many reports on the modification of the MR nucleus BNCz. Can the authors provide a detailed overview of the application of phosphine-containing materials in OLEDs to facilitate readers' understanding?
2. The authors are suggested to provide temperature-dependent transient PL decay spectra of the doped films to further support the TADF features.
3. In Table S4, the photophysical data of two compounds in doped films only have the ΔE_{ST} values. It is better to provide fluorescence and phosphorescence spectra of the doped films at 77 K.
4. In HF OLEDs using 5TCzBN as a TADF sensitizer, the performances of the doped OLEDs with higher concentrations (e.g., 15 wt% or more) should be further investigated to achieve optimal device performance.
5. In the device section, the authors should supplement the data of CE and PE at 100 and 1000 cd m⁻² to increase the completeness of device parameters.
6. As the purity of the material significantly impacts the performance of the device, the authors should provide purity information for both materials, such as HPLC data.
7. Page 9, Line 31 "...with a very narrow FWHM of ca. 20 nm for BNCz-NPO and BNCz-NPS..." The description of the FWHM is inconsistent with Table 1. Please double check it.

Reviewer #1:

In this paper, chemical modification of the MR emitter was tried using donating group. The concept of the paper is not novel because the same donor substitution was already tried in several papers. Although detailed analysis of the materials was carried out, the novelty of the material design concept is relatively low.

Comments:

Regarding the reference material BNCz, comparisons were made in some analyses, while other analyses did not include a comparative analysis. To better understand the improvements in the synthesized material, please provide a supplementary analysis of the areas that are not compared to promote a more effective understanding.

Response:

Thank you for your comprehensive review of our manuscript and for providing valuable feedback. We appreciate the opportunity to address your comments and clarify the unique contributions of our study.

Regarding the novelty of our work, while we acknowledge that chemical modification of MR emitters with electron-donating groups has been explored in previous literature, we emphasize that our study introduces a novel concept known as conformationally-flexible-donor-incorporation (CFDI). This strategy combines CT excited-state modulation and the SVC effect to address the crucial issue of low RISC rate in MR-TADF emitters, resulting in highly efficient pure-blue emitters with superior electroluminescence performance. Unlike conventional donor modifications, our CFDI approach preserves narrowband emission and luminescence efficiency, achieving state-of-the-art device performances. Additionally, our study elucidates the structure-property relationship of the new materials, providing insights into their design principles and potential applications. Notably, while the similar structures of NPO and NPS have been reported previously, their application as donors in electroluminescent materials has not been explored to the best of our knowledge.

As suggested, we enhanced our discussion by providing additional comparative analyses where applicable in the revised manuscript (for detailed information, please refer to the

response provided to the specific comment below), particularly in areas where comparisons with the control molecule BNCz were not previously made. This would offer a more comprehensive understanding of the advancements achieved with our new materials compared to the reference molecule.

Overall, we believe that our study offers novel insights into MR-TADF emitters, advancing the understanding of exciton dynamics and RISC processes. We appreciate your feedback and remain committed to furthering the knowledge and progress in this field.

Theoretical investigation

1. In Figure 3, the analysis of the S_n and T_n states in the excited state suggested that the RISC process could be improved. However, it should be incorporated the calculation results for the reference material to identify any differences from the reference material and elucidate them in the main text.

In addition, the DFT analysis indicates a substantial difference of 0.45 eV between S_1 and T_1 , which was significantly larger than the experimental value of 0.15 eV. Therefore, it is uncertain whether the SOC calculation results for the S_1 and T_n states asserted in Figure 3 are reasonable. Considering the ref.29 you mentioned in the introduction (Page 4, line 74), employing basis sets such as “SCS-CC2” in the DFT calculations could potentially yield more suitable results for MR-type materials.

(Ref. 29: J. Chem. Theory Comput. 2022, 593, 4903.)

Response:

Thank you for your valuable feedback and insightful suggestions.

We agree that the incorporation of calculation results for the reference material is essential to compare our findings and provide a more comprehensive analysis. In the revised manuscript, we have included the relevant calculation results for the reference material (BNCz) and discuss any discrepancies observed in the main text.

Furthermore, following your suggestion, we employed a new calculation method using SCS-CC2/cc-pVDZ to potentially yield more suitable predictions for our MR-TADF molecules. We also re-calculated the SOC matrix elements. We observed that the ΔE_{ST} values of BNCz,

BNCz-NPO, and BNCz-NPS are comparable. Additionally, we found that BNCz-NPO and BNCz-NPS indeed exhibit CT natures in high-lying excited states, which would lead to enhanced SOC for rapid RISC.

Taking into account the modifications suggested by other reviewers' comments, we have re-arranged the calculation section to incorporate the new results in the revised manuscript. We believe these updates will enhance the clarity and accuracy of our findings.

The revised statement is as follows (Page 8-9, highlighted):

“Figure 3a depicts the flexible potential energy curve scan of **BNCz-NPO** and **BNCz-NPS** at ground state. It is found that a shallow potential well, with an energy difference lower than thermal energy at room temperature ($k_B T \approx 0.026$ eV, k_B : Boltzmann constant, T : temperature), exists in a wide range of θ from -40 to 47° for **BNCz-NPO**. This range becomes larger from -45 to 50° for **BNCz-NPS** with higher conformational flexibility. Additionally, we found that the α conformer of **BNCz-NPO** (**BNCz-NPS**) exhibits a minimal potential energy difference compared to the β conformer, resulting in a nearly equal Boltzmann distribution ratio between them at room temperature (Figure 3b). This helps elucidate the observed polymorphism of **BNCz-NPO** and **BNCz-NPS**.

Excited state calculations were performed using the spin-component scaling second-order approximate coupled-cluster (SCS-CC2) method with the cc-pVDZ basis set.³⁰ Figure S7 illustrates a comparatively low ΔE_{ST} value of ~ 0.12 eV estimated in vacuum for the new compounds, similar to that of BNCz. Natural transition orbital (NTO) analyses were conducted to gain insights into the excited-state nature. Consistent with BNCz (Figure S8), the S_1 and T_1 states of **BNCz-NPO** and **BNCz-NPS** predominantly localize on their MR core (Figure 3cd), indicating LE characteristics (1LE and 3LE). In contrast, some higher-lying excited states, such as S_2 , T_2 , and T_4 , of **BNCz-NPO** and **BNCz-NPS** exhibit long-range CT or hybridized local and charge-transfer (HLCT) features, while those of BNCz remain dominated by LE. These CT characteristics are quantified by larger values of the distance of charge transfer (D_{CT}) and the amount of charge transferred (q_{CT}),⁴¹ as shown in Table S3. Due to the involvement of CT excited states distinct from the original LE-typed orbital feature of the MR core, both **BNCz-NPO** and **BNCz-NPS** display significantly larger SOC matrix elements than BNCz (Figure S9). Notably, **BNCz-NPS** exhibits larger D_{CT} and q_{CT} values than **BNCz-NPO**, resulting in larger

SOC matrix elements than **BNCz-NPO**. These SOC values notably surpass those of most MR-type TADF emitters.⁴²

Excited states with CT nature are typically more sensitive to the polarity of the environment compared to those with an LE nature. Thus, while **BNCz-NPO** and **BNCz-NPS** exhibit comparable excited state energy levels to BNCz due to the weak electron-donating nature of NPO/NPS, their S₂, T₂, and T₄ excited states become more stabilized than those of BNCz in polar CH₂Cl₂ (Figure S7). Additionally, due to subtle differences in excited-state energies among different conformers, **BNCz-NPO** and **BNCz-NPS** actually have a much richer landscape of excited states. Furthermore, the conformationally flexible NPO/NPS facilitates appropriate vibrational overlap between the nearly degenerate excited states, promoting reverse internal conversion. In this scenario, although the spatial orbital occupation (LE) of the S₁ and T₁ excitations results in vanishing direct SOC, the close-lying upper CT (or HLCT) excited states of **BNCz-NPO** and **BNCz-NPS** provide a feasible channel for efficient RISC, wherein higher-order SOC involving ³LE (or ³CT) and ¹CT (or ¹LE) in thermal equilibrium plays a crucial role.⁴³

Figure 3. a) Potential energy surface scans of the ground state of **BNCz-NPO** and **BNCz-NPS** under vacuum conditions. b) Boltzmann distributions of the geometries with different dihedral angles at room temperature. Hole-particle distribution for the excited singlet and triplet states of c) **BNCz-NPO** and d) **BNCz-NPS**.

Table S3 The distance of charge transfer (D_{CT}), and the amount of charge transferred (q_{CT}) in excited states

Compound	S ₁		S ₂		T ₁		T ₂		T ₃		T ₄	
	$D_{CT}/\text{\AA}$	q_{CT}	$D_{CT}/\text{\AA}$	q_{CT}	$D_{CT}/\text{\AA}$	q_{CT}	$D_{CT}/\text{\AA}$	q_{CT}	$D_{CT}/\text{\AA}$	q_{CT}	$D_{CT}/\text{\AA}$	q_{CT}
BNCz	1.04	0.57	0.78	0.54	0.62	0.59	1.15	0.42	1.02	0.42	0.58	0.49
BNCz-NPO	1.15	0.58	3.37	0.93	0.57	0.58	1.76	0.55	1.05	0.47	1.07	0.52
BNCz-NPS	1.18	0.61	3.65	0.95	0.58	0.60	1.44	0.64	1.24	0.53	3.64	0.92

Figure S7. Energy-level diagrams for the excited states of BNCz, BNCz-NPO, and BNCz-NPS in both vacuum and CH₂Cl₂ environments. Bar charts are used to illustrate the energy differences between these environments.

Figure S8. Hole-particle distribution for the excited singlet and triplet states of BNCz.

Figure S9. SOC heatmaps of BNCz, BNCz-NPO, and BNCz-NPS.

Photophysical properties

2. In Figure 4, a comprehensive comparative analysis with the reference material BNCZ facilitated a clear identification of improvements in the synthesized material. It is recommended to extend this comparison to the transient PL data in Figure 4b, providing an analysis of BNCZ's results for a more thorough understanding of the synthesized material.

Response:

Thank you for the valuable suggestion. We appreciate the opportunity to enhance the comparative analysis in our manuscript. In response to the reviewer's comment, we have included a more comprehensive comparison between the reference material BNCz and our synthesized materials, BNCz-NPO and BNCz-NPS, in the revised manuscript.

We have added the transient PL decay of BNCz in oxygen-free toluene solution to Figure

S13 (prompt fluorescence part) and Figure 4b (delayed fluorescence part). The PL decays of BNCz, BNCz-NPO, and BNCz-NPS consist of ns-scale prompt fluorescence and μ s-scale delayed fluorescence components. The corresponding prompt and delayed lifetimes were fitted to be 4.8 ns/13.9 μ s for BNCz-NPO and 4.7 ns/9.7 μ s for BNCz-NPS, respectively. The lifetimes of delayed components of BNCz-NPO and BNCz-NPS are dramatically reduced compared to that of BNCz (66.8 μ s), indicating a much more efficient RISC process.

The related discussion is also updated in the main text accordingly (Page 12, highlighted): “PL decays of BNCz, BNCz-NPO and BNCz-NPS consist of ns-scale prompt fluorescence (Figure S13) and μ s-scale delayed fluorescence components (Figure 4b). The corresponding prompt (τ_{PF}) and delayed (τ_{DF}) lifetimes were fitted to be 4.8 ns/13.9 μ s for BNCz-NPO and 4.7 ns/9.7 μ s for BNCz-NPS, respectively. In contrast, BNCz demonstrated a significantly longer τ_{DF} of 66.8 μ s, indicating a much more efficient RISC process in the presence of CFDI strategy.”

Figure 4. b) Transient PL decay curves of BNCz, BNCz-NPO and BNCz-NPS in oxygen-free toluene solution using a variable pulsed laser ($\lambda_{ex} = 375$ nm).

Figure S13. Transient PL decay curves of BNCz, BNCz-NPO and BNCz-NPS in oxygen-free toluene solution using a picosecond pulsed diode laser ($\lambda_{ex} = 375$ nm)

3. Transient PL data in Figure. 4b and the calculated constant rate in Table 1 fill in information about the solution state, while excepting the solid-state result to supporting information. Since the solid-state result tends to be more similar to the actual device, please transfer this data to the main text.

Response:

We appreciate the reviewer's insightful comment. As suggested, we put the PL data and calculated the constant rate parameters of solution and doped film to Table 1 in the main text accordingly. We believe that readers will access to a more detailed analysis of our findings.

Table 1. Photophysical data and kinetic parameters of BNCz, BNCz-NPO and BNCz-NPS in toluene (1×10^{-5} M) and doped films (3 wt% in PhCzBCz).

Emitter	State	λ_{abs}^a [nm]	λ_{em}^b [nm]	FWHM ^c [nm/eV]	E_{S1}^d [eV]	E_{T1}^d [eV]	ΔE_{ST}^e [eV]	$\Phi_{PF/DF}^f$ [%]	τ_{PF}^g [ns]	τ_{DF}^h [μs]	k_r^i [10^7 s ⁻¹]	k_{nr}^i [10^6 s ⁻¹]	k_{ISC}^i [10^7 s ⁻¹]	k_{RISC}^i [10^5 s ⁻¹]
BNCz	sol	468	483	23/0.13	2.62	2.52	0.10	76.4/16.8	5.4	66.8	14.1	10.3	3.3	0.18
	film	-	493	32/0.17	2.61	2.52	0.09	73.9/17.8	2.9	34.8	25.5	23.1	6.7	0.36
BNCz-NPO	sol	463	476	20/0.10	2.67	2.52	0.15	31.5/66.8	4.8	13.9	6.5	1.1	14.2	2.2
	film	-	478	26/0.14	2.66	2.56	0.10	20.0/75.5	2.8	28.5	7.1	3.3	28.2	1.6
BNCz-NPS	sol	462	475	21/0.11	2.64	2.50	0.14	25.3/66.5	4.7	9.7	5.3	4.8	15.4	3.7
	film	-	481	26/0.14	2.64	2.54	0.10	15.3/76.7	3.0	21.9	5.1	4.5	27.8	2.7

^aPeak of absorption spectrum. ^bPeak of fluorescence spectrum. ^cFull width at half-maximum (FWHM) of fluorescence spectrum. ^dLowest excited singlet (E_S) and triplet (E_T) energies estimated from peaks of the fluorescence and low-temperature phosphorescence spectra recorded at 77 K. ^e $\Delta E_{ST} = E_S - E_T$. ^fAbsolute photoluminescence quantum yield, Fractional quantum yields for prompt fluorescence (Φ_{PF}) and delayed fluorescence (Φ_{DF}). ^gLifetime of prompt fluorescence, ^hLifetime of delayed fluorescence. ⁱRate constants of singlet radiative decay (k_r), non-radiative decay (k_{nr}), intersystem crossing (k_{ISC}), reverse intersystem crossing (k_{RISC}).

4. Page 12. From line 270. “Importantly, compared to the small k_{RISC} of the prototypical BNCz ($1.34 \times 10^4 \text{ s}^{-1}$),⁴² those of BNCz-NPO and BNCz-NPS are markedly increased by 16.8 and 27.7 folds, reaching 2.25×10^5 and $3.71 \times 10^5 \text{ s}^{-1}$, respectively.”

The RISC rate constant of Ref. 42 is the result of the film state calculation, and the value of the synthesized material is compared and analyzed with the solution state. Please refer to the results analyzed in the same analysis condition to correct the degree of improvement in the RISC rate.

Response:

Thank you for bringing this to our attention. We have thoroughly reviewed the analysis conditions for comparing the RISC rate constants. Upon further investigation, we found that the k_{RISC} value from ref.42 (Mater. Horiz. 9, 2226–2232 (2022)) was obtained from a doped film with a different host material (SF3TRZ). To address this, we conducted additional measurements of the PL data for BNCz in both solution and film states, and these parameters are now listed in Table 1 for a comprehensive comparison, aligning well with your suggestions from comments 1 and 2. In the main text, we have revised the comparison to ensure that the RISC rate constants of the synthesized materials are appropriately analyzed with the solution state results. The revised comparison now accurately reflects the degree of improvement in the RISC rate. We believe these revisions enhance the accuracy and clarity of our manuscript.

The revised statement is as follows (Page 13, highlighted): “Importantly, compared to the small k_{RISC} of the prototypical BNCz ($1.8 \times 10^4 \text{ s}^{-1}$, see Table 1), those of **BNCz-NPO** and **BNCz-NPS** are increased by 12.2 and 20.5 folds, reaching 2.2×10^5 and $3.7 \times 10^5 \text{ s}^{-1}$, respectively.”

5. As you mentioned in introduction part, the addition of an additional donor group to the MR core as a way to improve RISC rate may lead to a decrease in PLQY (Page 3. From line 62), so please compare and mention the results of PLQY analysis between the synthesized material and the reference material in the main text.

Response:

Thank you for your valuable suggestion. We have taken this point into careful consideration and made the necessary revisions to address it. Expressly, we have incorporated the PLQY values of the reference compound BNCz and compared them with our newly synthesized emitters in the revised manuscript. These PLQY values are now included in the updated Table 1 ($\text{PLQY} = \Phi_{\text{PF}} + \Phi_{\text{DF}}$).

The revised statement is as follows (Page 10, highlighted): “The PLQYs of BNCz-NPO and BNCz-NPS in dilute toluene solution are as high as 98.3% and 91.8%, respectively, which are comparable to that of BNCz (93.2%).”

Table 1. Photophysical data and kinetic parameters of BNCz, **BNCz-NPO** and **BNCz-NPS** in toluene (1×10^{-5} M) and doped films (3 wt% in PhCzBCz).

Emitter	State	$\lambda_{\text{abs}}^{\text{a}}$	$\lambda_{\text{em}}^{\text{b}}$	FWHM ^c	E_{S1}^{d}	E_{T1}^{d}	$\Delta E_{\text{ST}}^{\text{e}}$	$\Phi_{\text{PF/DF}}^{\text{f}}$	$\tau_{\text{PF}}^{\text{g}}$	$\tau_{\text{DF}}^{\text{h}}$	k_{r}^{i}	k_{nr}^{i}	$k_{\text{ISC}}^{\text{i}}$	$k_{\text{RISC}}^{\text{i}}$
		[nm]	[nm]	[nm/eV]	[eV]	[eV]	[eV]	[%]	[ns]	[μs]	[10^7 s ⁻¹]	[10^6 s ⁻¹]	[10^7 s ⁻¹]	[10^5 s ⁻¹]
BNCz	sol	468	483	23/0.13	2.62	2.52	0.10	76.4/16.8	5.4	66.8	14.1	10.3	3.3	0.18
	film	-	493	32/0.17	2.61	2.52	0.09	73.9/17.8	2.9	34.8	25.5	23.1	6.7	0.36
BNCz-	sol	463	476	20/0.10	2.67	2.52	0.15	31.5/66.8	4.8	13.9	6.5	1.1	14.2	2.2
NPO	film	-	478	26/0.14	2.66	2.56	0.10	20.0/75.5	2.8	28.5	7.1	3.3	28.2	1.6
BNCz-	sol	462	475	21/0.11	2.64	2.50	0.14	25.3/66.5	4.7	9.7	5.3	4.8	15.4	3.7
NPS	film	-	481	26/0.14	2.64	2.54	0.10	15.3/76.7	3.0	21.9	5.1	4.5	27.8	2.7

^aPeak of absorption spectrum. ^bPeak of fluorescence spectrum. ^cFull width at half-maximum (FWHM) of fluorescence spectrum. ^dLowest excited singlet (E_{S}) and triplet (E_{T}) energies estimated from peaks of the fluorescence and low-temperature phosphorescence spectra recorded at 77 K. ^e $\Delta E_{\text{ST}} = E_{\text{S}} - E_{\text{T}}$. ^fAbsolute photoluminescence quantum yield, Fractional quantum yields for prompt fluorescence (Φ_{PF}) and delayed fluorescence (Φ_{DF}). ^gLifetime of prompt fluorescence, ^hLifetime of delayed fluorescence. ⁱRate constants of singlet radiative decay (k_{r}), non-radiative decay (k_{nr}), intersystem crossing (k_{ISC}), reverse intersystem crossing (k_{RISC}).

There are only ΔE_{ST} values in the main text and table, and no S_1 , T_1 data. For intuitive confirmation, please add singlet, triplet energy values.

Response:

Thank you for your valuable suggestion. In the revised manuscript, singlet/triplet energy values were added to the main text and Table 1 accordingly.

The revised statement is as follows (Page 12, highlighted): “From the fluorescence and phosphorescence spectra in toluene measured at 77 K (Figure S12), the S_1 and T_1 energy levels are estimated to be 2.67/2.52 and 2.64/2.50 eV for **BNCz-NPO** and **BNCz-NPS**, respectively. Subsequently, the ΔE_{ST} s of **BNCz-NPO** and **BNCz-NPS** are determined to be 0.15 and 0.14 eV, respectively, which align well with the calculation results.”

Table 1. Photophysical data and kinetic parameters of **BNCz**, **BNCz-NPO** and **BNCz-NPS** in toluene (1×10^{-5} M) and doped films (3 wt% in PhCzBCz).

Emitter	State	λ_{abs}^a	λ_{em}^b	FWHM ^c	E_{S1}^d	E_{T1}^d	ΔE_{ST}^e	$\Phi_{PF/DF}^f$	T_{PF}^g	T_{DF}^h	k_r^i	k_{nr}^i	k_{ISC}^i	k_{RISC}^i
		[nm]	[nm]		[nm/eV]	[eV]	[eV]				[eV]	[10 ⁷ s ⁻¹]	[10 ⁶ s ⁻¹]	[10 ⁷ s ⁻¹]
BNCz	sol	468	483	23/0.13	2.62	2.52	0.10	76.4/16.8	5.4	66.8	14.1	10.3	3.3	0.18
	film	-	493	32/0.17	2.61	2.52	0.09	73.9/17.8	2.9	34.8	25.5	23.1	6.7	0.36
BNCz-NPO	sol	463	476	20/0.10	2.67	2.52	0.15	31.5/66.8	4.8	13.9	6.5	1.1	14.2	2.2
	film	-	478	26/0.14	2.66	2.56	0.10	20.0/75.5	2.8	28.5	7.1	3.3	28.2	1.6
BNCz-NPS	sol	462	475	21/0.11	2.64	2.50	0.14	25.3/66.5	4.7	9.7	5.3	4.8	15.4	3.7
	film	-	481	26/0.14	2.64	2.54	0.10	15.3/76.7	3.0	21.9	5.1	4.5	27.8	2.7

^a)Peak of absorption spectrum. ^b)Peak of fluorescence spectrum. ^c)Full width at half-maximum (FWHM) of fluorescence spectrum. ^d)Lowest excited singlet (E_S) and triplet (E_T) energies estimated from peaks of the fluorescence and low-temperature phosphorescence spectra recorded at 77 K. ^e) $\Delta E_{ST} = E_S - E_T$. ^f)Absolute photoluminescence quantum yield, Fractional quantum yields for prompt fluorescence (Φ_{PF}) and delayed fluorescence (Φ_{DF}). ^g)Lifetime of prompt fluorescence, ^h)Lifetime of delayed fluorescence. ⁱ)Rate constants of singlet radiative decay (k_r), non-radiative decay (k_{nr}), intersystem crossing (k_{ISC}), reverse intersystem crossing (k_{RISC}).

Figure S12. Fluorescence (blue line) and phosphorescence (red line) spectra of a) BNCz, b) BNCz-NPO and c) BNCz-NPS in toluene solution (1×10^{-5} M, 77 K); Fluorescence (blue line) and phosphorescence (red line) spectra of a) BNCz, b) BNCz-NPO and c) BNCz-NPS in doped films (3 wt% in PhCzBCz).

Crystallographic analysis

7. Page 7. From line 161. “Moreover, the bulky NPO/NPS, with significant steric hindrance, induces a nearly orthogonal conformation with an MR-NPO/NPS dihedral angle exceeding 80° effectively suppressing strong interchromophore interactions (Figure S4).”

As a result of single crystal analysis, it was said that the introduction of NPO/NPS units into the core would inhibit the intermolecular interaction. In order to confirm the characteristics of the unit, please analysis and explain how effectively excimer and concentration quenching were suppressed through analysis such as solid-PL, transient PL, PLQY according to the doping concentration along with the reference core.

Response:

Thank you for your insightful comment. We appreciate the opportunity to provide further analysis and clarification regarding the anti-quenching effect induced by the NPO/NPS modification. In response to your suggestions, we conducted additional experiments and analyses to investigate the characteristics of the modified units and their impact on intermolecular interactions and solid-state luminescence properties.

Firstly, to delve deeper into the impact of NPO/NPS modification on intermolecular interactions, we employed Hirshfeld surface analysis to scrutinize the packing mode in single crystals. As depicted in Figure S5, the red isosurface portion, indicative of intermolecular contacts, is predominantly distributed on the NPO/NPS fragments and *tert*-butyl groups of the MR backbone. The full fingerprint plot reveals that **BNCz-NPO** and **BNCz-NPS** exhibit sparser contact densities and larger contact distances than **BNCz**, suggesting weaker intermolecular interactions. Additionally, by plotting the decomposed fingerprint for the MR core to analyze π - π interactions before and after NPO/NPS modification. The involvement of planar stacking arrangements for **BNCz** is evident as a red region (increased density) near the center of the plot ($1.8 \text{ \AA} < d_{i,e} < 2.0 \text{ \AA}$). Notably, the interchromophore contact density is greatly reduced in **BNCz-NPO** and **BNCz-NPS**, indicating alleviated interchromophore interaction.

Figure S5. (a) Hirshfeld surface. The red isosurface portion, indicative of intermolecular contacts, is predominantly distributed on the NPO/NPS fragments and *tert*-butyl groups of the MR backbone. (b) Single molecule full fingerprint plot. The full fingerprint plot reveals that BNCz-NPO and BNCz-NPS exhibit sparser contact densities and larger contact distances than BNCz, suggesting weaker intermolecular interactions. (c) decomposed fingerprint plots for specific pairs of the C-C contacts on the MR core. The involvement of planar stacking

arrangements for BNCz is evident as a red region (increased density) near the center of the plot ($1.8 \text{ \AA} < d_{i,e} < 2.0 \text{ \AA}$). In contrast, the interchromophore contact density is greatly reduced in **BNCz-NPO** and **BNCz-NPS**, indicating alleviated interchromophore interaction.

Subsequently, we prepared PhCzBCz-hosted films using **BNCz-NPO**, **BNCz-NPS**, or **BNCz** as a dopant with different doping levels and systematically measured their photophysical properties (see Figure S16). It is revealed that **BNCz-NPO** and **BNCz-NPS** displayed reduced excimer emission and were less susceptible to concentration-induced emission quenching compared to **BNCz**. Additionally, the lifetimes of **BNCz-NPO** and **BNCz-NPS** exhibited slower decreasing trends with increasing doping concentrations, indicative of their superior solid-state luminescence properties.

Figure S16. Photophysical properties of **BNCz**, **BNCz-NPO** and **BNCz-NPS** as a dopant in **PhCzBCz**-hosted film with different doping levels. (a) PL spectra. As the dopant level increased from 1 to 30 wt%, the PL spectra of **BNCz-NPO**- and **BNCz-NPS**-based doped films exhibited a relatively stable profile with a slight redshift. This contrasts sharply with the emergence of excimer emission observed in **BNCz**. (b) Plots for the FWHM and PLQY versus dopant concentration. Compared to **BNCz**, **BNCz-NPO** and **BNCz-NPS** exhibited suppressed concentration-induced emission quenching and spectral broadening, maintaining small FWHM values and relatively high PLQYs within the 1–30 wt% concentration range. (c) Transient PL decay curves. The lifetime of **BNCz** gradually decreased with increasing doping concentrations, while the decrease trends were much slower for **BNCz-NPO**- and **BNCz-NPS**.

Furthermore, we investigated the PL behaviors in THF/water mixtures with varying water fractions to highlight the capability to mitigate aggregation-caused quenching (ACQ). The results showed that while **BNCz** exhibited typical ACQ behavior with increasing water fractions, **BNCz-NPO** and **BNCz-NPS** maintained stable emission intensities and FWHM values,

indicating noteworthy anti-quenching characteristics imparted by the NPO/NPS functionalization (Figure S17).

Figure S17. Photophysical properties of (a) BNCz, (b) BNCz-NPO and (c) BNCz-NPS in THF/water mixtures with different water fractions (f_w). Left panel: PL spectra in THF/water mixtures with different f_w ; right panel: plots of the relative PL intensity (I/I_0) and full-width at half-maximum (FWHM) of PL spectra versus f_w in a THF/water mixtures. Concentration: 10 μM .

These findings underscore the effectiveness of the NPO/NPS modification in suppressing intermolecular interactions and mitigating ACQ, thereby enhancing the solid-state

luminescence properties of MR-TADF emitters. These observations are consistent with the insights gleaned from crystallographic and computational analyses. We believe that these additional analyses and experimental results further strengthen the robustness and significance of our study.

The revised statements are as follows:

“We employed Hirshfeld surface analysis to scrutinize the packing mode in single crystals of **BNCz-NPO** and **BNCz-NPS** (Figure S5). The analysis revealed sparser contact densities and larger contact distances compared to BNCz, indicating weaker intermolecular interactions and reduced interchromophore contacts.” (Page 7, highlighted);

“The investigation of photophysical properties for our new emitters extended to various doping levels (Figure S16). The findings revealed that **BNCz-NPO** and **BNCz-NPS** displayed reduced excimer emission and were less susceptible to concentration-induced emission quenching compared to BNCz. Additionally, the lifetimes of **BNCz-NPO** and **BNCz-NPS** exhibited slower decreasing trends with increasing doping concentrations, indicative of their superior solid-state luminescence properties. Furthermore, we investigated the PL behaviors in THF/water mixtures with varying water fractions to highlight the capability to mitigate aggregation-caused quenching (ACQ), as shown in Figure S17. The results showed that while BNCz exhibited typical ACQ behavior with increasing water fractions, **BNCz-NPO** and **BNCz-NPS** maintained stable emission intensities and FWHM values, indicating noteworthy anti-quenching characteristics imparted by the NPO/NPS functionalization. These observations are consistent with the insights gleaned from crystallographic and computational analyses.”

(Page 13, highlighted)

Electroluminescence Properties

8. Page 15. From line 376.

In the Hyperfluorescence OLED system, the introduction of an interlayer sensitization structure was mentioned to suppress Dexter energy transfer and further optimize the system. However, it would be beneficial to provide more detailed quantification or explanation, such as specific values related to exciton lifetimes or the degree to which Dexter energy transfer is inhibited for

a more understanding.

Response:

Thank you for your insightful comment regarding the HF-OLEDs discussed in our study. We appreciate the opportunity to address this clarification. Upon reviewing the original manuscript, we recognize an inaccuracy in our explanation. The optimized performance of the interlayer sensitization structure in HF-OLEDs is not solely due to the suppression of Dexter energy transfer, as previously stated. Instead, it is primarily attributed to the more efficient RISC of the TADF sensitizer doped in the high-polarity host, which alleviates triplet-involved exciton quenching.

In our previous studies (Z. Zhao et al. *Adv. Mater.* 2023, 35, 2212237), we identified that hosts suitable for MR-TADF emissive species are typically of low polarity, such as the PhCzBCz used in the manuscript and commonly used mCBP. However, a challenge arises when TADF sensitizers are incorporated into low-polarity hosts, often resulting in weak TADF properties. To address this issue, we proposed an interlayer sensitization device (Z. Zhao et al. *Adv. Funct. Mater.* 2021, 31, 2103273), which divides the structure into two separated layers: MR-TADF emissive species doped into low-polarity hosts and blue TADF sensitizers doped into high-polarity hosts. In this way, the TADF properties of the blue TADF sensitizers are enhanced, facilitating sensitizing efficiency via long-range Förster energy transfer (FET), and ultimately improving the performance of the sensitization device.

In the revised manuscript, this error has been rectified, and the relevant references have been cited accordingly. We appreciate your attention to detail and apologize for any confusion caused by the oversight.

The revised statements are as follows (**Page 17, highlighted**):

“Though the efficiency roll-offs are significantly suppressed, the improvement of maximum EQE in HF-I OLEDs are not obvious. Thus, further optimization was performed using an interlayer sensitization structure⁵³ with a more efficient TADF sensitizer, 9-(5'-(4,6-diphenyl-1,3,5-triazin-2-yl) [1,1':3',1''-terphenyl]-2'-yl)-3,6-diphenyl-9H-carbazole (PPCz-Trz).⁵⁴ Another set of HF OLEDs was fabricated with a device configuration of ITO/HATCN (5 nm)/TAPC (50 nm)/TCTA (5 nm)/3 wt% **BNCz-NPO (BNCz-NPS)**: PhCzBCz (10 nm)/10 wt% PPCz-TRZ: PPF (2 nm)/PPF (5 nm)/TmPyPB (30 nm)/LiF (1 nm)/Al (120 nm) (HF-II).

The device structure and characteristics are shown in Figure S24. In HF-II OLEDs, the PPCz-Trz sensitizer was doped into PPF with high polarity to harvest exciton energy more efficiently, thereby enhancing sensitizing efficiency through a long-range Förster energy transfer process.⁵⁵

9. Furthermore, the authors should provide device lifetime results as a benchmark for future research.

Response:

Thank you for your valuable suggestion regarding the inclusion of device lifetime results in our study. We acknowledge the importance of providing a benchmark data for future research. In response to your comment, we have conducted device lifetime testing and have included the results in the revised manuscript.

Figure S21 illustrates the operational lifetime of the **BNCz-NPO**- and **BNCz-NPS**-based OLEDs, with an initial luminance of 100 cd m⁻² (display-relevant). We observed that the **BNCz-NPO**-based OLED exhibits a significantly prolonged *T*₅₀ (defined as the time when the brightness diminishes to half of its initial value), defined as the time when the luminance diminishes to half of its initial value, surpassing three-fold that of the **BNCz-NPS**-based device (15.2 hrs compared to 3.0 hrs).

While our primary focus remains on the molecular design for high luminescence efficiency and color purity, the observed difference in device lifetimes between **BNCz-NPO** and **BNCz-NPS** OLEDs may provide insights into the impact of molecular structure on operational stability. We speculate that the lower EL stability of **BNCz-NPS** could be attributed to the weaker bonding of the P=S group in its molecular structure.

It is worth mentioning that the realization of operationally stable blue OLEDs remains a challenging issue across the field. In the future, we will continue to contribute to the ongoing efforts to address this challenge based on the insights gained from our findings in this study.

The revised statements are as follows (Page 16, highlighted): “Figure S21 illustrates EL stabilities of **BNCz-NPO** and **BNCz-NPS**. We observed that the **BNCz-NPO**-based OLED exhibits a significantly prolonged *T*₅₀ (defined as the time when the brightness diminishes to half of its initial value), defined as the time when the luminance diminishes to half of its initial

value, surpassing three-fold that of the **BNCz-NPS**-based device (15.2 hrs compared to 3.0 hrs). We speculate that the lower EL stability of **BNCz-NPS** could be attributed to the weaker bonding of the P=S group in its molecular structure.”

Figure SX. Operation lifetimes of the **BNCz-NPO** and **BNCz-NPS** OLEDs at an initial luminance of 100 cd m^{-2} . Device structure: ITO/MoO₃ (6.5 nm)/Tris-PCz (20 nm)/mCBP (20 nm)/3 wt% **BNCz-NPO** or **BNCz-NPS** : mCBP (20 nm)/SF3-TRZ (10 nm)/50 wt% SF3-TRZ: Liq (30 nm)/LiF (1 nm)/Al (120 nm).

Reviewer #2 (Remarks to the Author):

In this manuscript, the authors proposed a conformation-flexible-donor-incorporation (CFDI) strategy, which can simultaneously regulate the long-range charge transfer (CT) character and spin-vibronic coupling (SVC) of molecules. Based on this strategy, two efficient blue luminescent materials, BNCz-NPO and BNCz-NPS, were designed based on a MR-TADF molecule BNCZ by introducing NPO and NPS as functional donors. Surprisingly, the transition characteristics of the S1 and T1 states of the molecules do not be changed after the introduction of functional donor, and NPO and NPS do not generate more vibration modes in the low frequency region. As a result, the fluorescence spectra were not broadened. The FWHM values

of BNCz-NPO and BNCz-NPS in toluene are only 20 nm and 21 nm. The functional donors NPO and NPS equip the higher energy excited states (S₂, T₂, etc.) with CT nature, which allows the spin flipping process. Moreover, the flexible structure of the donor group increases the number of triple excited state nearby the S₁ state, which increases the intense SVC-mediated RISC channels and greatly improves the kRISC value. The device employing BNCz-NPO (BNCz-NPS) as the emitter exhibited a pure blue emission with an FWHM of 26 nm (30 nm) and an EQEmax of 37.6% (32.2%). These results are excellent. The manuscript is recommended for publication with minor revisions presented below.

Response:

We thank the reviewer for these positive comments supporting the novelty of our work and the significance of our claims in the manuscript. We hope that our revision will clarify your concerns.

1. The single crystal analysis of BNCz-NPO and BNCz-NPS indicated two stable configurations in both crystals. I wonder which configuration the molecules tend to stay in solution or film. Can the two configurations transform each other?

Response:

Thank you for your insightful question regarding the molecular configurations observed in our single-crystal analysis. We appreciate the opportunity to provide further clarification on this matter.

In our study, the single crystal analysis revealed the presence of two stable configurations for both **BNCz-NPO** and **BNCz-NPS**. However, it is important to acknowledge that the behavior of molecules in solution or film may differ from that in the crystalline state due to the influence of various factors such as solvation effects, intermolecular interactions, and packing arrangements. While our investigation did not specifically address the transformation between the two configurations in solution or film, it is conceivable that dynamic processes such as conformational changes or molecular rearrangements could occur under specific conditions, potentially leading to the interconversion of configurations.

We conducted theoretical calculations to evaluate the conformational variability of **BNCz-NPO** and **BNCz-NPS** to gain further insights into this issue. Figure 3a illustrates the flexible potential energy curve scan of both compounds at the ground state. The analysis revealed the existence of a shallow potential well within a broad range of dihedral angles, indicating appreciable conformational distributions at room temperature. Additionally, we found that the α conformer of **BNCz-NPO** (**BNCz-NPS**) exhibits a minimal potential energy difference compared to the β conformer, resulting in a nearly equal Boltzmann distribution ratio between them at room temperature (Figure 3b). This finding helps elucidate the observed polymorphism of **BNCz-NPO** and **BNCz-NPS**.

We believe that these theoretical insights complement our experimental findings and provide a deeper understanding of the conformational dynamics of **BNCz-NPO** and **BNCz-NPS**.

The revised statements are as follows (Page 8, highlighted): “Figure 3a depicts the flexible potential energy curve scan of **BNCz-NPO** and **BNCz-NPS** at the ground state. It is found that a shallow potential well, with an energy difference lower than thermal energy at room temperature ($k_B T \approx 0.026$ eV, k_B : Boltzmann constant, T : temperature), exists in a wide range of θ from -40 to 47° for **BNCz-NPO**. This range becomes larger from -45 to 50° for **BNCz-NPS** with higher conformational flexibility. Additionally, we found that the α conformer of **BNCz-NPO** (**BNCz-NPS**) exhibits a minimal potential energy difference compared to the β conformer, resulting in a nearly equal Boltzmann distribution ratio between them at room temperature (Figure 3b). This helps elucidate the observed polymorphism of **BNCz-NPO** and **BNCz-NPS**.”

Figure 3. a) Potential energy surface scans of the ground state of BNCz-NPO and BNCz-NPS under vacuum conditions. b) Boltzmann distributions of the geometries with different dihedral angles at room temperature. Hole-particle distribution for the excited singlet and triplet states of c) BNCz-NPO and d) BNCz-NPS.

2. The PLQYs of BNCz-NPO and BNCz-NPS in toluene are given in Table 1. Please also give

the PLQYs of fluorescence and TADF components.

Response:

We appreciate the valuable suggestion from the reviewer. In response, we have revised Table 1 in the manuscript to incorporate the PLQYs in toluene accordingly. We have also included the PLQY values of fluorescence and TADF components.

Table 1. Photophysical data and kinetic parameters of BNCz, BNCz-NPO and BNCz-NPS in toluene (1×10^{-5} M) and doped films (3 wt% in PhCzBCz).

Emitter	State	$\lambda_{\text{abs}}^{\text{a}}$	$\lambda_{\text{em}}^{\text{b}}$	FWHM ^c	$E_{\text{S}_1}^{\text{d}}$	$E_{\text{T}_1}^{\text{d}}$	$\Delta E_{\text{ST}}^{\text{e}}$	$\Phi_{\text{PF/DF}}^{\text{f}}$	$\tau_{\text{PF}}^{\text{g}}$	$\tau_{\text{DF}}^{\text{h}}$	k_{r}^{i}	k_{nr}^{i}	$k_{\text{ISC}}^{\text{i}}$	$k_{\text{RISC}}^{\text{i}}$
		[nm]	[nm]	[nm/eV]	[eV]	[eV]	[eV]	[%]	[ns]	[μs]	[10^7 s ⁻¹]	[10^6 s ⁻¹]	[10^7 s ⁻¹]	[10^5 s ⁻¹]
BNCz	sol	468	483	23/0.13	2.62	2.52	0.10	76.4/16.8	5.4	66.8	14.1	10.3	3.3	0.18
	film	-	493	32/0.17	2.61	2.52	0.09	73.9/17.8	2.9	34.8	25.5	23.1	6.7	0.36
BNCz-NPO	sol	463	476	20/0.10	2.67	2.52	0.15	31.5/66.8	4.8	13.9	6.5	1.1	14.2	2.2
	film	-	478	26/0.14	2.66	2.56	0.10	20.0/75.5	2.8	28.5	7.1	3.3	28.2	1.6
BNCz-NPS	sol	462	475	21/0.11	2.64	2.50	0.14	25.3/66.5	4.7	9.7	5.3	4.8	15.4	3.7
	film	-	481	26/0.14	2.64	2.54	0.10	15.3/76.7	3.0	21.9	5.1	4.5	27.8	2.7

^aPeak of absorption spectrum. ^bPeak of fluorescence spectrum. ^cFull width at half-maximum (FWHM) of fluorescence spectrum. ^dLowest excited singlet (E_{S}) and triplet (E_{T}) energies estimated from peaks of the fluorescence and low-temperature phosphorescence spectra recorded at 77 K. ^e $\Delta E_{\text{ST}} = E_{\text{S}} - E_{\text{T}}$. ^fAbsolute photoluminescence quantum yield, Fractional quantum yields for prompt fluorescence (Φ_{PF}) and delayed fluorescence (Φ_{DF}). ^gLifetime of prompt fluorescence, ^hLifetime of delayed fluorescence. ⁱRate constants of singlet radiative decay (k_{r}), non-radiative decay (k_{nr}), intersystem crossing (k_{ISC}), reverse intersystem crossing (k_{RISC}).

3. The energy levels and NTOs of Sn and Tm states are given in Figure 3. However, the NTO analysis should involve a transition from one orbital to another, such as NTO No. 15 to NTO No. 16, rather than showing only one NTO in the figure. Additionally, the BNCz molecule is used as a reference in the paper. Please provide the calculation results of BNCz, including the energy-levels and NTOs of the Sn and Tm, and the SOC matrix elements between the Tm and Sn.

Response:

Thank you for your valuable suggestions. In response to your first point regarding the NTO analysis, we have updated Figure 3 to include transitions from one orbital to another. In addition, we have included the relevant calculation results for the reference molecule, BNCz (see Figure S7,S8,S9). These additional calculations allow for a direct comparison between our MR-TADF molecules and the non-modified BNCz, providing a clearer understanding of our findings.

The revised statements are as follows (Page 8, highlighted): “Excited state calculations were performed using the spin-component scaling second-order approximate coupled-cluster (SCS-CC2) method with the cc-pVDZ basis set.³⁰ Figure S7 illustrates a comparatively low ΔE_{ST} value of ~ 0.12 eV estimated in vacuum for the new compounds, similar to that of BNCz. Natural transition orbital (NTO) analyses were conducted to gain insights into the excited-state nature. Consistent with BNCz (Figure S8), the S_1 and T_1 states of **BNCz-NPO** and **BNCz-NPS** predominantly localize on their MR core (Figure 3cd), indicating LE characteristics (1LE and 3LE). In contrast, some higher-lying excited states, such as S_2 , T_2 , and T_4 , of **BNCz-NPO** and **BNCz-NPS** exhibit long-range CT or hybridized local and charge-transfer (HLCT) features, while those of BNCz remain dominated by LE. These CT characteristics are quantified by larger values of the distance of charge transfer (D_{CT}) and the amount of charge transferred (q_{CT}),⁴¹ as shown in Table S3. Due to the involvement of CT excited states distinct from the original LE-typed orbital feature of the MR core, both **BNCz-NPO** and **BNCz-NPS** display significantly larger SOC matrix elements than BNCz (Figure S9). Notably, **BNCz-NPS** exhibits larger D_{CT} and q_{CT} values than **BNCz-NPO**, resulting in larger SOC matrix elements than **BNCz-NPO**. These SOC values notably surpass those of most MR-type TADF emitters.⁴²

Excited states with CT nature are typically more sensitive to the polarity of the environment compared to those with an LE nature. Thus, while **BNCz-NPO** and **BNCz-NPS** exhibit comparable excited state energy levels to BNCz due to the weak electron-donating nature of NPO/NPS, their S_2 , T_2 , and T_4 excited states become more stabilized than those of BNCz in polar CH_2Cl_2 (Figure S7). Additionally, due to subtle differences in excited-state energies among different conformers, **BNCz-NPO** and **BNCz-NPS** actually have a much richer landscape of excited states. Furthermore, the conformationally flexible NPO/NPS facilitates appropriate vibrational overlap between the nearly degenerate excited states, promoting reverse internal conversion. In this scenario, although the spatial orbital occupation (LE) of the S_1 and

T_1 excitations results in vanishing direct SOC, the close-lying upper CT (or HLCT) excited states of **BNCz-NPO** and **BNCz-NPS** provide a feasible channel for efficient RISC, wherein higher-order SOC involving 3LE (or 3CT) and 1CT (or 1LE) in thermal equilibrium plays a crucial role.⁴³

Figure 3. a) Potential energy surface scans of the ground state of **BNCz-NPO** and **BNCz-NPS** under vacuum conditions. b) Boltzmann distributions of the geometries with different dihedral

angles at room temperature. Hole-particle distribution for the excited singlet and triplet states of c) **BNCz-NPO** and d) **BNCz-NPS**.

Figure S8. Hole-particle distribution for the excited singlet and triplet states of BNCz.

Figure S9. SOC heatmaps of BNCz, BNCz-NPO, and BNCz-NPS.

Figure S7. Energy-level diagrams for the excited states of BNCz, BNCz-NPO, and BNCz-NPS in both vacuum and CH_2Cl_2 environments. Bar charts are used to illustrate the energy differences between these environments.

Reviewer #3 (Remarks to the Author):

This manuscript presents a design strategy for blue MR-TADF materials by incorporating conformationally flexible donor (NPO and NPS) moieties to enhance the spin-flip of the molecules while maintaining narrowband emission. The novel and effective molecular design

make these two molecules exhibit outstanding device performance in both non-sensitized and sensitized devices, placing the molecules at the forefront of the MR-TADF field. The proposed design strategy challenges conventional thinking by effectively integrating flexible and rigid groups, offering readers a fresh approach to molecular design. I recommend publication of the work in Nature Communications after minor revisions as outlined below:

Response:

We thank you for these positive comments supporting the novelty of our work and the significance of our claims in the manuscript. We hope that our revision will clarify your concerns.

1. There have been many reports on the modification of the MR nucleus BNCz. Can the authors provide a detailed overview of the application of phosphine-containing materials in OLEDs to facilitate readers' understanding?

Response:

Thank you for your insightful comment. While the modification of the MR core BNCz has indeed been explored in various studies, the application of phosphine-containing materials in OLEDs represents an important area of research that warrants further elucidation. In response to this valuable feedback, we have revised the molecular design section to provide additional context and clarity, thereby enhancing readers' understanding.

The revised statement now reads as follows (Page 5, highlighted): “A prototypical MR framework, BNCz, was employed as the light-emitting core due to its excellent optical properties.³⁵ Despite several proposed modification strategies to enhance its performance, most BNCz derivatives emit in the blue-green spectrum, presenting a challenge to simultaneously achieving efficient RISC and pure-blue emission. Herein, we introduce an NPO/NPS unit onto the *para*-position of the BNCz skeleton with respect to the B atom, to yield two new MR-TADF emitters, namely **BNCz-NPO** and **BNCz-NPS**, are depicted in Figure 1. Compared to previously reported modifying building blocks for MR cores, NPO/NPS serves as a versatile excited-state modifier with distinct advantages. The NPO/NPS unit exhibits moderate-to-weak electron-donating ability due to the negative inductive effect of the embedded P=O/P=S (refer to Figure S1).”

2. The authors are suggested to provide temperature-dependent transient PL decay spectra of the doped films to further support the TADF features.

Response:

Thank you for your insightful suggestion. We have conducted additional experiments to measure the temperature-dependent transient PL decay spectra of the doped films (Figure S14). These spectra provide valuable insights into the TADF properties of the materials across different temperature ranges. The results of these experiments have been included in the revised manuscript to further support the TADF features observed in our study.

Figure S14. Temperature-dependent transient PL decays of a) BNCz-NPO and b) BNCz-NPS in oxygen-free toluene solution and Temperature-dependent transient PL decays of c) BNCz-NPO and d) BNCz-NPS in doped films (3 wt% in PhCzBCz) using a variable pulsed laser (Edinburgh VPL-375).

3. In Table S4, the photophysical data of two compounds in doped films only have the ΔEST values. It is better to provide fluorescence and phosphorescence spectra of the doped films at 77 K.

Response:

We appreciate your insightful comment. In response to your comment, we have conducted additional measurements to include the fluorescence and phosphorescence spectra of the doped films at 77 K for BNCz, BNCz-NPO and BNCz-NPS (Figure S12d-f). These spectra will provide a more complete understanding of the photophysical properties of the materials in the doped film state. The updated data will be included in Table 1 (Table S4 in the original manuscript) of the revised manuscript accordingly.

**Figure S12**. Fluorescence (blue line) and phosphorescence (red line) spectra of a) BNCz, b) **BNCz-NPO** and c) **BNCz-NPS** in toluene solution ($1 \times 10^{-5}$ M, 77 K); Fluorescence (blue line) and phosphorescence (red line) spectra of d) BNCz, e) **BNCz-NPO** and f) **BNCz-NPS** in doped films (3 wt% in PhCzBCz).

**Table 1.** Photophysical data and kinetic parameters of BNCz, **BNCz-NPO** and **BNCz-NPS** in toluene ($1 \times 10^{-5}$ M) and doped films (3 wt% in PhCzBCz).

| Emitter | State | $\lambda_{abs}$[a] [nm] | $\lambda_{em}$[b] [nm] | FWHM[c] [nm/eV] | $E_{S1}$[d] [eV] | $E_{T1}$[d] [eV] | $\Delta E_{ST}$[e] [eV] | $\Phi_{PF/DF}$[f] [%] | $\tau_{PF}$[g] [ns] | $\tau_{DF}$[h] [μs] | $k_r$[i] [$10^7$ | $k_{nr}$[i] [$10^6$ | $k_{ISC}$[i] [$10^7$ | $k_{RISC}$[i] [$10^5$ |
|---|---|---|---|---|---|---|---|---|---|---|---|---|---|---|

										s ⁻¹]	s ⁻¹]	s ⁻¹]	s ⁻¹]	
BNCz	sol	468	483	23/0.13	2.62	2.52	0.10	76.4/16.8	5.4	66.8	14.1	10.3	3.3	0.18
	film	-	493	32/0.17	2.61	2.52	0.09	73.9/17.8	2.9	34.8	25.5	23.1	6.7	0.36
BNCz-	sol	463	476	20/0.10	2.67	2.52	0.15	31.5/66.8	4.8	13.9	6.5	1.1	14.2	2.2
NPO	film	-	478	26/0.14	2.66	2.56	0.10	20.0/75.5	2.8	28.5	7.1	3.3	28.2	1.6
BNCz-	sol	462	475	21/0.11	2.64	2.50	0.14	25.3/66.5	4.7	9.7	5.3	4.8	15.4	3.7
NPS	film	-	481	26/0.14	2.64	2.54	0.10	15.3/76.7	3.0	21.9	5.1	4.5	27.8	2.7

^a)Peak of absorption spectrum. ^b)Peak of fluorescence spectrum. ^c)Full width at half-maximum (FWHM) of fluorescence spectrum. ^d) Lowest excited singlet (E_S) and triplet (E_T) energies estimated from peaks of the fluorescence and low-temperature phosphorescence spectra recorded at 77 K. ^e) $\Delta E_{ST} = E_S - E_T$. ^f)Absolute photoluminescence quantum yield, Fractional quantum yields for prompt fluorescence (Φ_{PF}) and delayed fluorescence (Φ_{DF}). ^g)Lifetime of prompt fluorescence, ^h)Lifetime of delayed fluorescence. ⁱ) Rate constants of singlet radiative decay (k_r), non-radiative decay (k_{nr}), intersystem crossing (k_{ISC}), reverse intersystem crossing (k_{RISC}).

4. In HF OLEDs using 5TCzBN as a TADF sensitizer, the performances of the doped OLEDs with higher concentrations (e.g., 15 wt% or more) should be further investigated to achieve optimal device performance.

Response:

Thank you for your insightful suggestion. In response to your comment, we have conducted further investigations into the performances of the doped OLEDs with higher 5TCzBN concentrations of 15 and 20 wt%. The results of these additional experiments have been included in the revised manuscript (Figure S23 and Table S6). We can see that the optimal concentration of 5TCzBN is 10 wt%.

Figure S23. a) Device structure of HF- I type OLEDs based on **BNCz-NPO** and **BNCz-NPS**, with different doping ratio (10 wt%, 15 wt%, and 20 wt%) of 5TCzBN. b) EL spectra measured at 100 cd m⁻². c) current density-voltage-luminance characteristics. d) EQE-luminance curves.

Table S6. Summary of the HF- I type OLEDs performances based on **BNCz-NPO** and **BNCz-NPS**.

Emitter	Conc. of 5TCzBN	V _{on} ^{a)}	L _{max} ^{b)}	CE ^{c)}	PE ^{d)}	EQE ^{e)}	λ _{EL} ^{f)}	FWHM	CIE ^{g)}
	[wt%]	[V]	[cd m ⁻²]	[cd A ⁻¹]	[lm W ⁻¹]	[%]	[nm]	^{g)} [nm]	[x, y]
BNCz-NPO	10 wt%	2.8	22660	40.6/37.5/27.6	45.5/37.7/23.3	30.2/28.0/20.7	480	26	(0.12, 0.22)
BNCz-NPO	15 wt%	2.8	22230	38.2/37.4/28.6	40.0/37.6/23.4	27.9/27.3/20.8	480	26	(0.12, 0.22)
BNCz-NPO	20 wt%	2.8	27220	36.6/35.5/28.7	41.1/36.8/25.0	26.4/25.5/20.6	478	26	(0.12, 0.22)
BNCz-NPS	10 wt%	3.0	26330	50.1/49.4/41.5	52.4/50.3/36.1	31.9/31.6/26.4	480	30	(0.13, 0.24)
BNCz-NPS	15 wt%	2.8	24580	50.4/49.5/44.1	52.7/51.8/38.2	29.4/29.2/25.6	478	32	(0.14, 0.26)
BNCz-NPS	20 wt%	2.8	32410	51.4/50.1/43.8	57.7/53.0/40.0	30.7/29.7/25.9	478	32	(0.13, 0.24)

^{a)} Voltage measured at 1 cd m⁻²; ^{b)} maximum luminance; ^{c)} maximum current efficiency at maximum, 100, 1000 cd m⁻², respectively; ^{d)} maximum power efficiency at maximum, 100, 1000 cd m⁻², respectively; ^{e)} external quantum efficiency at maximum, 100, 1000 cd m⁻², respectively; ^{f)} EL peak; ^{g)} full width at half maximum of EL spectrum; ^{h)} recorded at 4 V.

5. In the device section, the authors should supplement the data of CE and PE at 100 and 1000 cd m^{-2} to increase the completeness of device parameters.

Response:

Thank you for your valuable suggestion. We have supplemented the data of current efficiency and power efficiency at both 100 and 1000 cd m^{-2} in the device section of the revised manuscript (see updated Table 2).

Table 2. Summary of the EL performances based on **BNCz-NPO** and **BNCz-NPS**.

Emitter	Conc. [wt%]	$V_{\text{on}}^{\text{a)}$ [V]	$L_{\text{max}}^{\text{b)}$ [cd m^{-2}]	CE ^{c)} [cd A^{-1}]	PE ^{d)} [lm W^{-1}]	EQE ^{e)} [%]	$\lambda_{\text{EL}}^{\text{f)}$ [nm]	FWHM ^{g)} [nm]	CIE ^{h)} [x, y]
BNCz-NPO	1	3.2	10510	32.6/23.8/13.0	32.0/21.0/9.1	30.4/22.2/12.1	478	25	(0.11, 0.17)
BNCz-NPO	3	3.0	10960	43.0/28.5/13.1	45.0/26.4/9.5	32.1/21.7/9.7	480	26	(0.11, 0.22)
BNCz-NPO	5	3.0	10170	40.5/28.4/11.3	42.4/26.3/8.3	27.8/19.5/7.8	480	26	(0.11, 0.25)
BNCz-NPO (HF-I)	3	2.8	22660	40.6/37.5/27.6	45.5/37.7/23.3	30.2/28.0/20.7	480	26	(0.12, 0.22)
BNCz-NPO (HF-II)	3	2.8	13850	44.7/36.2/14.1	50.1/37.4/12.7	37.6/30.2/12.0	480	26	(0.11, 0.18)
BNCz-NPS	1	3.2	4906	23.7/16.7/9.0	23.3/14.3/6.1	28.5/20.0/10.8	474	26	(0.12, 0.14)
BNCz-NPS	3	3.2	6491	30.6/24.4/14.0	30.0/22.0/10.1	29.6/23.5/13.5	476	26	(0.11, 0.16)
BNCz-NPS	5	3.2	6154	30.3/25.8/14.1	29.7/23.9/10.4	27.4/23.3/12.6	478	26	(0.11, 0.17)
BNCz-NPS (HF-I)	3	3.0	26330	50.1/49.4/41.5	52.4/50.3/36.1	31.9/31.6/26.4	480	30	(0.14, 0.24)
BNCz-NPS (HF-II)	3	3.0	14640	40.2/37.0/21.9	39.5/34.4/17.4	32.2/29.2/17.3	476	30	(0.13, 0.18)

^{a)} Voltage measured at 1 cd m^{-2} ; ^{b)} maximum luminance; ^{c)} maximum current efficiency at maximum, 100, 1000 cd m^{-2} , respectively; ^{d)} maximum power efficiency at maximum, 100, 1000 cd m^{-2} , respectively; ^{e)} external quantum efficiency at maximum, 100, 1000 cd m^{-2} , respectively; ^{f)} EL peak; ^{g)} full width at half maximum of EL spectrum; ^{h)} recorded at 4 V.

6. As the purity of the material significantly impacts the performance of the device, the authors should provide purity information for both materials, such as HPLC data.

Response:

We appreciate the reviewer's insightful comment. To address the concern raised, we conducted high-performance liquid chromatography (HPLC) to assess the purities of both BNCz-NPO and BNCz-NPS after the sublimation purification process. The results, presented in **Figure S43**, confirm that the compounds purified via temperature-gradient vacuum sublimation exhibit purities exceeding 99%.

Figure S43. HPLC spectra monitored at 254 nm with THF-water ratio of 65/35 (v/v) for the sources of **BNCz-NPO** and **BNCz-NPS**.

7. Page 9, Line 31“...with a very narrow FWHM of ca. 20 nm for BNCz-NPO and BNCz-NPS...” The description of the FWHM is inconsistent with Table 1. Please double check it.

Response:

Thank you for bringing this inconsistency to our attention. Upon careful examination, we have identified and corrected the typographical error accordingly.

The revised statement is as follows (Page 10, highlighted): “It is worth noting that the PL spectra of the new MR-TADF emitters are unstructured, with a very narrow FWHM of 20 and 21 nm for **BNCz-NPO** and **BNCz-NPS**, respectively.”

REVIEWERS' COMMENTS

Reviewer #1 (Remarks to the Author):

Main concern about this paper is the novelty of the material design differentiating this work from other works. Authors tried to rationalize the concept of the paper. However, the concern about the chemical design concept could not be resolved in the revised paper.

Reviewer #2 (Remarks to the Author):

The paper has been adequately revised and can be recommended for publication.

Reviewer #3 (Remarks to the Author):

OK now.